# SEMI-SUPERVISED CONSISTENCY REGULARIZATION FOR ACCURATE CELL TYPE FRACTION AND GENE EXPRESSION ESTIMATION

## ABSTRACT

Cell deconvolution is the estimation of cell type fractions and cell type-specific gene expression from mixed data with unknown composition. In biomedical research, cell deconvolution, which is a source separation task, is used to obtain mechanistic and diagnostic insights into human diseases. An unmet challenge in cell deconvolution, however, is the scarcity of realistic training data and the strong domain shift observed in synthetic training data that is used in contemporary methods. Here, we hypothesize that simultaneous consistency regularization of the target and training domains will improve deconvolution performance. By adding this biologically motivated consistency loss to two novel deep learning-based deconvolution algorithms, we achieve state-of-the-art performance on both cell fraction and gene expression estimation. Our method, DISSECT, outperforms competing algorithms across several gene expression datasets and can be easily adapted to deconvolve other biomedical data types, as exemplified by our spatial expression deconvolution experiments.

## 1 INTRODUCTION

A prominent approach to study tissue-specific gene expression changes in human development and disease is RNA sequencing (bulk RNA-seq). Tissues, however, usually consist of multiple cell types in different quantities, with different gene expression programs. As a consequence, bulk RNA-seq from tissues measures average gene expression across the constituent cells, disregarding cell type-specific changes. The quantification of the cellular composition and cell type-specific expression that underlies bulk RNA-seq data is therefore of pivotal importance to understand disease mechanisms and to identify potential therapeutic interventions (Li & Wang, 2021).

A recent technological advancement, single-cell RNA-seq, allows for the investigation of gene expression in single cells for thousands of individual cells of a given tissue sample in a single experiment. While it provides unprecedented insights into single cell biology, it suffers from severe technical limitations, most notably gene expression 'dropouts' (Lähnemann et al., 2020). In addition, the technology is still very costly, which largely prohibits its application in clinical and diagnostic settings. Bulk RNA-seq, on the other hand, can be performed for a fraction of the cost and is widely used in clinical oncology and drug discovery (Zhou et al., 2019; Roberts et al., 2014).

To computationally infer cell fraction and cell type-specific gene expression information from bulk RNA-seq data, recent computational methods utilize single cell sequencing data to create simulated references with known fraction and expression information for training (Avila Cobos et al., 2020; Menden et al., 2020; Newman et al., 2019; Wang et al., 2019). While this approach achieves good deconvolution results, its performance suffers from the strong domain gap between single-cell RNA-seq training (reference) data and the bulk RNA-seq target data. Among many possible sources of variation, two most obvious are the presence of batch effects which refers to technological differences between two sequencing experiments and gene expression differences of biological nature. In the next section we formally define the task of cell deconvolution, and present our hypothesis that semi-supervised consistency regularization minimizes the bulk RNA-seq deconvolution error while learning from single cell RNA-seq data.

## 2 CELL DECONVOLUTION

Given an $m \times n$ gene expression matrix $\mathbf{B}$ consisting of $m$ bulk gene expression vectors measuring $n$ genes, the goal of deconvlution is to find a $m \times c$ matrix $\mathbf{X}$ of cell type fractions, where $c$ is the number of cell types present in bulk samples such that,

$$\mathbf{B} = \mathbf{XS}, \tag{1}$$

where fractions and gene expression satisfy non-negativity $0 \leq \mathbf{X}_{ik}$, and $0 \leq \mathbf{S}_{kj}, \forall i \in [1, m], \forall j \in [1, n]$ and $\forall k \in [1, c]$ and sum-to-1 criterion *i.e.* $\sum_{k=1}^{c} \mathbf{X}_{ik} = 1, \forall i \in [1, m]$. Here, $\mathbf{S}$ is known as the signature matrix and is unobserved. Each row $\mathbf{S}_{k\cdot}$ is a gene expression profile (or signature) of cell type $k$. To utilize a reference based framework, $\mathbf{S}$ can be replaced with $\mathbf{S}_{ref}$ derived from a single-cell experiment.

The problem of reference-based cell deconvolution can alternatively be formulated as a learning problem, where a function $f$ such that $f(\mathbf{B}) = \mathbf{X}$ is learnt. Since only $\mathbf{B}$ is available and $\mathbf{X}$ is generally unknown, simulations from single-cell reference can be used to learn $f$. Clearly, from the above formulation of the cell deconvolution task, it is reasonable to assume linearity of deconvolution, i.e., each bulk mixture is a linear combination of expression vectors of cells spanned with corresponding cell type fractions. Thus, as defined in (Menden et al., 2020), multiple single cells can be combined in random proportions to generate training examples $\mathbf{B}^{\text{sim}}$ and $\mathbf{X}^{\text{sim}}$, where each row of $\mathbf{B}^{\text{sim}}$ is defined as,

$$\mathbf{B}_{i\cdot}^{\text{sim}} = \sum_{k=1}^{c} \sum_{l=1}^{\alpha_{k,i}} \mathbf{e}_{l}^{k},$$

where $\mathbf{e}_{l}^{k}$ is expression vector of cell $l$ belonging to cell type $k$, and $\alpha_{k,i}$ is the number of cells belonging to cell type $k$ sampled to construct $\mathbf{B}_{i\cdot}^{\text{sim}}$. Correspondingly, each element of $\mathbf{X}^{\text{sim}}$ is proportion of a cell type $k$ in that sample $i$ and is defined as,

$$\mathbf{X}_{ik}^{\text{sim}} = \frac{\alpha_{k,i}}{\sum_{k=1}^{c} \alpha_{k,i}}, \text{ and}$$

In this case, since each simulated sample has a distinct signature (i.e. gene expression profile), $\mathbf{S}$ is a three dimensional matrix with each element $\mathbf{S}_{kji}$ denoting gene expression of gene $j$ in cell type $k$ for sample $i$. It is computed as following,

$$\mathbf{S}_{k \cdot i}^{\text{sim}} = \frac{\sum_{l=1}^{\alpha_{k,i}} \mathbf{e}_{l}^{k}}{\alpha_{k,i}}.$$

The predictor $f$, learned from a simulated dataset, can then be applied to $\mathbf{B}$ to estimate $\mathbf{X}$. Note that, the genes expressed may differ between vectors $\mathbf{e}_l$ and $\mathbf{B}$ and as such before learning function $f$, each $\mathbf{e}_l^k$ is subsetted to include genes common with $\mathbf{B}$. This is the reason why this learning problem is transductive and a separate model needs to be reconstructed for each $\mathbf{B}$.

### 2.1 EXPLOITING LINEARITY OF DECONVOLUTION

#### 2.1.1 ASSUMPTION

From Section 2, it is evident that the relationship between $\mathbf{B}$ and $\mathbf{S}$ is linear. However, $\mathbf{S}$ is unobserved and learning is done using simulations. To address the inherent domain shift, we hypothesize that a consistency based regularization penalizing non-linearity of mixtures of real and simulated samples would result in a mapping $f$ that is closer to true $f$. We define it in Section 2.1.2.

### 2.1.2 CONSISTENCY REGULARIZATION

Consider $\mathbf{B}$ represents gene expression matrices of real (*i.e.* test) bulk RNA-seq that we want to deconvolve and and $\mathbf{B}^{\text{sim}}$ represents gene expression matrix of simulated bulk samples. The number of rows (representing samples) in these two matrices may differ. To simplify the notation, we use the same index $i$ for real bulk samples, simulations (sim) and their mixtures (mix, defined further). Given a true bulk RNA-seq $\mathbf{B}_{i\cdot}$, and a simulated sample with paired proportions $(\mathbf{B}_{i\cdot}^{\text{sim}}, \mathbf{X}_{i\cdot}^{\text{sim}})$ defined over common gene-set, we can generate a mixture $\mathbf{B}_{i\cdot}^{\text{mix}}$ such that

$$\mathbf{B}_{i\cdot}^{\text{mix}} = \beta\mathbf{B}_{i\cdot} + (1-\beta)\mathbf{B}_{i\cdot}^{\text{sim}}, \tag{2}$$

Which gives us relation

$$\mathbf{X}_{i\cdot}^{\text{mix}}\mathbf{S}_{\cdot i\cdot}^{\text{mix}} = \beta\mathbf{X}_{i\cdot}\mathbf{S}_{\cdot i\cdot} + (1-\beta)\mathbf{X}_{i\cdot}^{\text{sim}}\mathbf{S}_{\cdot i\cdot}^{\text{sim}}. \tag{3}$$

Cell types are characterized by a few marker genes that are invariant across cell states and even across tissues (Domínguez Conde et al., 2022). A network that accurately predicts cell type fractions based on gene expression of simulated (or real) bulks would thus have to learn them. Thus, to estimate cell type fractions, we assume that the expression of these marker genes should be identical in signatures $\mathbf{S}_{\cdot i\cdot}^{\text{mix}}, \mathbf{S}_{\cdot i\cdot}$ and $\mathbf{S}_{\cdot i\cdot}^{\textbf{sim}}$. Hence,

$$\mathbf{X}_{i\cdot}^{\text{mix}} = \beta\mathbf{X}_{i\cdot} + (1-\beta)\mathbf{X}_{i\cdot}^{\text{sim}}, \tag{4}$$

where $\beta \in [0,1]$. Equation 4 enables the use of consistency regularization without having to explicitly estimate signatures. In an iterative learning process $\mathbf{X}_{i\cdot}$ can be replaced with predictions of the algorithm from the previous iteration. Naturally, it is also possible to only mix real samples with each other, however, the number of samples available from true bulk RNA-seq is considerably lesser (usually ranging from a couple to less than thousand) than the amount single-cells present in a single-cell experiment (usually in thousands). The equation 4 allows to generate pseudo ground truth proportions for mixtures $\mathbf{B}_{i\cdot}^{\text{mix}}$ at the each step of learning cell type fractions, while Equation 3 allows to generate pseudo ground truth signatures at each step of learning gene expression profiles. We define the network architecture and loss functions in 2.2.

## 2.2 NETWORK ARCHITECTURE AND LEARNING PARADIGM

We approach the two tasks, estimation of cell type fractions and estimation of gene expression profiles per cell type as two different tasks because of their differing assumptions. For estimation of cell type fractions, we assume that signatures are identical for each sample, both simulated and bulk, while to estimate gene expression, we relax this condition use full consistency regularization (Equation 3).

### 2.2.1 ESTIMATION OF CELL TYPE FRACTIONS

The underlying algorithm of the first part of our deconvolution method is an average ensemble of multilayered perceptrons (MLPs). Each MLP consists of the same architecture initialized with different weights. This is done to reduce the variance by averaging different runs (Ju et al., 2018). Each MLP has an architecture: Input (# genes) - ReLU6 (512) - ReLU6 (256) - ReLU6 (128) - ReLU6 (64) - Linear (# cell types) - Softmax. ReLU6 (Output of ReLU activation clipped by a maximum value of 6) (Hannun et al., 2014; Sandler et al., 2018) was chosen out of tested activations over grid search on [Linear, ReLU, ReLU6, Swish (Ramachandran et al., 2017)] The final application of Softmax activation allows to achieve non-negativity and sum to 1 criteria of deconvolution. we train the network with batch size 64 to minimize the loss function defined below with an Adam Optimizer with initial learning rate of $1e-5$.

$$\mathcal{L}_{\text{total}}(\mathbf{X}_{i\cdot}^{\text{sim}}, f(\mathbf{B}_{i\cdot}^{\text{sim}}), \mathbf{X}_{i\cdot}^{\text{mix}}, f(\mathbf{B}_{i\cdot}^{\text{mix}})) = \mathcal{L}_{\text{KLdivergence}}(\mathbf{X}_{i\cdot}^{\text{sim}}, f(\mathbf{B}_{i\cdot}^{\text{sim}})) + \lambda_1 * \mathcal{L}_{\text{cons}}(\mathbf{X}_{i\cdot}^{\text{mix}}, f(\mathbf{B}_{i\cdot}^{\text{mix}})), \tag{5}$$

where $\mathcal{L}_{\text{KLdivergence}}(\cdot,\cdot)$ is the Kullback-Leibler divergence and $\mathcal{L}_{\text{cons}}(\cdot,\cdot)$ is the consistency loss defined as:

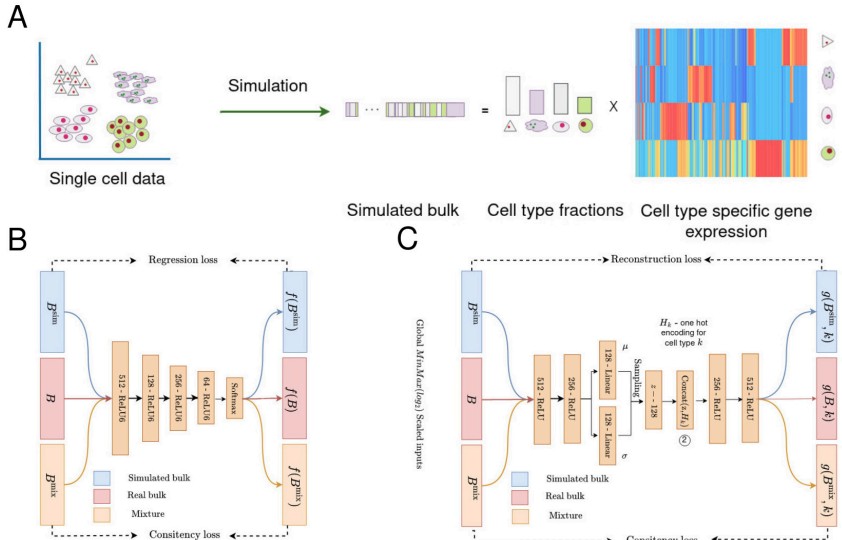

Figure 1: **A.** Illustration of simulation using reference single-cell data. The figure shows the simulation of one sample which consists of cell type fractions, simulated gene expression and cell type specific gene expression profiles (*i.e.* signature matrix). **B.** Detailed overview of an MLP used to estimate cell type fractions, and **C.** Overview of an autoencoder used to estimate cell type specific gene expression profiles.

$$\mathcal{L}_{\text{cons}}(\mathbf{X}_{i\cdot}^{\text{mix}}, f(\mathbf{B}_{i\cdot}^{\text{mix}})) = ||\mathbf{X}_{i\cdot}^{\text{mix}} - f(\mathbf{B}_{i\cdot}^{\text{mix}})||_2^2, \text{ and}$$

$$\mathbf{X}_{i\cdot}^{\text{mix}} = \beta f(\mathbf{B}_{i\cdot}) + (1 - \beta)\mathbf{X}_{i\cdot}^{\text{sim}}.$$

To generate mixtures, for each batch, we sample $\beta$ and uniformly at random for Equation 4. The interval $[0.1, 0.9]$ was chosen for the uniform distribution to allow for at least some real and some simulated gene expression in the mixture. This loss is similar to the semi-supervised framework proposed in MixMatch (Berthelot et al., 2019). MixMatch uses unlabelled samples to MixUp and match sample predictions and generalizes semi-supervised framework, while the loss defined in Equation 5 addresses the limited samples available from true bulk RNA-seq, unavailability of sample fractions and is derived from the definition of task itself. In essence, Equation 5 integrates domain knowledge into the objective.

To avoid a scenario where the network doesn't learn and outputs predictions such that $f(\mathbf{B}_{i\cdot}^{\text{mix}}) = f(\mathbf{B}_{i\cdot}^{\text{sim}}) = f(\mathbf{B}_{i\cdot})$, which is a solution to Equation 4, we first let the model learn purely from simulated examples. This allows the model to learn meaningful expression profiles to achieve accurate results on simulated examples. We selected $\lambda_1$ based on a grid search over constant and step-wise functions. We adopt a step-wise function for $\lambda_1$, given as:

$$\lambda_1 = \begin{cases} 0 & \text{if step} \leq 2000, \\ 15 & \text{elif } 2000 \leq \text{step} \leq 4000, \\ 10 & \text{else.} \end{cases}$$

We train the network for a predefined number of steps as opposed to epochs, since it is possible to generate infinitely many simulated samples without increasing the intrinsic dimensionality of the data. In our experiments, we set their number to 5000 as found optimal in Menden et al. (2020).

### 2.2.2 ESTIMATION OF PER SAMPLE CELL TYPE SPECIFIC GENE EXPRESSION PROFILES

Estimation of cell type fractions from bulk RNA-seq requires an assumption that signatures of cell types are shared across single cell and bulk RNA-seq. However, cell type gene expression profiles

(at least for genes that are not invariant across tissue states) may differ between samples. Previously, works such as CSx (Newman et al., 2019) and TAPE (Chen et al., 2021) have explored utilizing cell type fractions to estimate gene expression per sample. Here, we make use of a $\beta$-variational autoencoder with standard normal distribution as prior to estimate average gene expression of the different cell types from bulk RNA-seq expression levels. To jointly train the network on all cell types, we condition the decoder (at its input layer) with cell type labels. This allows for training a single model to estimate gene expression of each cell type for a sample. To make use of bulk RNA seq during the training, we regularize the reconstruction loss with a consistency loss defined over per cell type signature. Denoting $f$ as before and $g(\cdot, k)$ as the output of the autoencoder with condition $k$ (corresponding to cell type label) on the decoder input, this consistency loss is defined as:

$$\mathcal{L}_{\text{cons}}^{\text{VAE}}(f, g, \mathbf{B}_{i\cdot}^{\text{mix}}, \mathbf{B}_{i\cdot}, \mathbf{X}_{i\cdot}^{\text{sim}}, \mathbf{S}_{ki\cdot}^{\text{sim}}) = ||f(\mathbf{B}_{i\cdot}^{\text{mix}})_k g(\mathbf{B}_{i\cdot}^{\text{mix}}, k) - \beta f(\mathbf{B}_{i\cdot})_k g(\mathbf{B}_{i\cdot}, k) + (1-\beta)\mathbf{X}_{i\cdot}^{\text{sim}}\mathbf{S}_{ki\cdot}^{\text{sim}}||_2^2,$$

where $\mathbf{B}_i^{\text{mix}}$ is given by Equation 2, $f(\mathbf{B}_{i\cdot}^{\text{mix}})_k$ is the estimated proportion of cell type $k$ in sample $i$. In implementation, we replace $f(\mathbf{B}_{i\cdot}^{\text{mix}})_k$ with $\beta f(\mathbf{B}_{i\cdot})_k + (1 - \beta)\mathbf{X}_{i\cdot}^{\text{sim}}$. Thus, this loss forces the learned signature for cell type $k$, $g(\mathbf{B}_{i\cdot}^{\text{mix}}, k)$, to be closer to signatures for both real and simulated bulk samples. This loss function makes the assumption that mixing two bulk samples is similar to mixing individual cell type specific signatures that constitute those bulks. We added this loss function with a regularization parameter $\lambda_2$ (with default value 0.1) to the loss of the standard $\beta$-variational autoencoder (the weight on the KL divergence, denoted as $\beta^{\text{VAE}}$, is set to 0.1 by default). The total loss function sums up to:

$$\begin{aligned}\mathcal{L}_{\text{total}}^{\text{VAE}}(f, g, \mathbf{B}_{i\cdot}^{\text{sim}}, \mathbf{B}_{i\cdot}^{\text{mix}}, \mathbf{B}_{i\cdot}, \mathbf{X}_{i\cdot}^{\text{sim}}, \mathbf{S}_{ki\cdot}^{\text{sim}}) &= ||\mathbf{S}_{ki\cdot}^{\text{sim}} - g(\mathbf{B}_{i\cdot}^{\text{sim}}, k)||_2^2 \\ &+ \lambda_2 \mathcal{L}_{\text{cons}}^{\text{VAE}}(f, g, \mathbf{B}_{i\cdot}^{\text{mix}}, \mathbf{B}_{i\cdot}, \mathbf{X}_{i\cdot}^{\text{sim}}, \mathbf{S}_{ki\cdot}^{\text{sim}}) \\ &+ \beta^{VAE} \mathcal{L}_{\text{KLdivergence}}(\mathcal{N}(\mu, \sigma), \mathcal{N}(0, 1)),\end{aligned}$$

where $\mathcal{N}(0, 1)$ is standard normal distribution, $\mu$ and $\sigma$ are the empirical mean and standard deviation estimated from the output of the encoder. Both the encoder and decoder consist of two hidden layers. We train the network to minimize the loss function with an Adam optimizer with initial learning rate of $1e - 4$. By default, the network is trained with a batch size of 32 for $10000 \times c$ number of steps. The architecture of the network is summarized in Figure 1.

## 3 RELATED WORK AND COMPETING ALGORITHMS

Several methods for cell deconvolution have been developed. Avila Cobos et al. (2020) and Jin & Liu (2021) provided a benchmark and review of state of the art cell deconvolution algorithms. Here, we focus on MuSiC, CSx, Scaden and TAPE although here are several additional methods such as DWLS (Tsoucas et al., 2019) and Bisque (Jew et al., 2020) etc., because both MuSiC and CSx are single cell reference based methods that have performed well on simulation studies in aforementioned benchmarking studies. Scaden and TAPE are selected as both are deep learning based deconvolution approaches. We briefly detail these approaches below. Out of these methods, CSx and TAPE can also estimate per sample cell type-specific gene expression signatures.

**MuSiC** (Wang et al., 2019) uses weighted non-negative least squares. MuSiC maintains cross-cell and cross-sample consistencies by appropriately weighting genes based on their informativity during an iterative procedure. MuSiC is provided as an R package. Deconvolution using MuSiC was performed according to the authors recommendations. Since MuSiC is a method that utilizes multi-subject scRNA-seq datasets, when available, we used cells from multiple subjects in deconvolution with MuSiC. **CibersortX** (CSx) (Newman et al., 2019) is a deconvolution method that addresses domain gap problems with scRNA-seq and bulk samples by aiming to correct batch effects. It uses scRNA-seq to generate a cell type specific signature matrix and uses $\nu$-support vector regression as the underlying algorithm. CSx comprises two modes, S- and B-modes, to address the domain gap. S-mode is used when deconvolving with a signature matrix constructed using a scRNA-seq dataset, while B-mode is used when deconvolving with a signature matrix constructed using purified samples. We followed the documentation provided by the authors to run CSx and used the S-mode. CSx

can also predict gene expression signatures for each sample for which it uses a Non-negative matrix factorization based iterative algorithm. **Scaden** (Menden et al., 2020) is an average ensemble of three deep neural networks with different architectures that was developed for cell fraction deconvolution. Each network is trained only on simulated pseudo bulk data generated from an scRNA-seq reference similar to described above. Scaden is provided as a Python package. We used the official Scaden package with the instructions provided by the authors to train the networks. **TAPE** (Chen et al., 2021) is a fully-connected autoencoder where the bottleneck consists of cell type fractions. The architecture of the encoder is similar to the archictecture of Scaden but with CeLU activations. The decoder consists of linear activations and outputs gene expression of the input vector. The adaptive mode of TAPE aims at optimizing the network for bulk samples, while the overall mode trains for fractions with an added loss function that reconstructs input bulk expression from fractions. Since TAPE-A reconstructs gene expression from fractions (bottleneck), the signature matrix is visible in the (linear) decoder. To estimate gene expression signatures for each bulk sample, decoder weights are optimized per-sample using an iterative optimization strategy. Network weights are changed during the two modes, we compare with both and refer to TAPE in overall mode as TAPE-O and in adaptive mode as TAPE-A. **Linear MLPs:** The solution to the deconvolution problem could be, in principle, a linear function. For this reason we also compared to an MLP ensemble that is based on the architecture in Section 2.2, but in which we replaced all non-linear activations with an identity function and removed the consistency loss.

## 4 EXPERIMENTS AND RESULTS

### 4.1 DATASETS

We evaluated the algorithms on six datasets consisting of peripheral blood mononuclear cells (PBMCs) and corresponding ground-truth cell type fractions experimentally quantified using flow cytometry. Details of these datasets are given in Table 1.

Table 1: Real bulk datasets with known ground truth cell fraction information for the evaluation of deconvolution performance.

| Dataset name | # samples | # genes | Original Source | Type |
|---|---|---|---|---|
| SDY67 | 12 | 11328 | Zimmermann et al. (2016) | RNA-seq |
| Monaco bulk | 12 | 17487 | Monaco et al. (2019) | RNA-seq |
| Monaco microarray | 164 | 38593 | Monaco et al. (2019) | Microarray |
| GSE65133 | 20 | 11328 | Newman et al. (2015) | Microarray |
| GSE107572 | 9 | 19423 | Finotello et al. (2019) | RNA-seq |
| GSE120502 | 250 | 20343 | Harrison et al. (2019) | RNA-seq |

To deconvolve these datasets, we used the *PBMC8k* (Table 2) as a reference single-cell dataset from a healthy donor for all methods considered here. To maintain same genes between the single-cell data and bulk RNA-seq, we subset both datasets over common gene-set. Number of common genes between *PBMC8k* and each of the bulk samples are as follows: *SDY67*: 10717, *Monaco bulk*: 13122, *Monaco microarray*: 13467, *GSE65133*: 10717, *GSE107572*: , *GSE120502*: 13699.

To deconvolve with deep learning based methods (Scaden, TAPE-O, TAPE-A, Linear MLPs and DISSECT), we use this single-cell data from healthy donors to create simulated bulk data with known fractions as described in section 2. The non-deep learning methods (MuSiC and CibersortX) do not require simulations and as such single-cell data is used without simulations. For the estimation of cell type specific gene expression per sample, we utilized simulations in the absence of corresponding ground truth in the real bulk RNA-seq. For this, in addition to *PBMC8k*, we considered three other reference datasets, namely *PBMC6k*, *donorA* and *donorC* (Table 2). The results are given in Section 4.3.

Since the number of cell types is unknown apriori in a bulk RNA-seq dataset that we want to deconvolve, we create an "unknown" cell type label in the reference dataset by merging cells not belonging to any of the cell types present in corresponding tissue (Menden et al., 2020; Chen et al., 2021). This unknown cluster allows comparison of fractions measured at relative scale. PBMCs consist of five main cell types namely CD4 T cells, CD8 T cells, NK cells, Monocytes and B cells (Bittersohl &

Table 2: Single-cell data used for the creation of simulated reference data for supervised training. We manually annotated cells with these five cell types based on expression of cell type marker genes. QC refers to Quality Control step (Appendix C.3) performed to filter out poor quality cells and genes that are expressed in too few cells.

| Dataset name | # cells | # genes (Pre-QC) | # genes (Post-QC) | Title in 10X Genomics[1] |
|---|---|---|---|---|
| PBMC8k | 8381 | 32738 | 14343 | 8k PBMCs from a Healthy Donor |
| PBMC6k | 5419 | 33694 | 18340 | 6k PBMCs from a Healthy Donor |
| DonorA | 2900 | 32738 | 13067 | Frozen PBMCs (Donor A) |
| DonorC | 9519 | 32738 | 15275 | Frozen PBMCs (Donor C) |

Steimer, 2016). Thereby, we end up with six cell types (including unknown cell type). Similarly, for the bulk RNA-seq datasets (Table 1), we grouped the ground truth cell type proportions not belonging to these five cell types in a single label "Unknown" following the methodology in Menden et al. (2020).

To preprocess single-cell datasets, we utilized the procedure described in Appendix C which includes quality control (QC) and simulations. A detailed information on on the parameters used for simulations are provided in Appendix C.2.

For the *Monaco bulk* dataset (Table 1), more granular level fractions of cell type subsets are quantified using flow cytometry. We utilized this information to evaluate methods on discerning closely related or scarce cell type subsets. Since it is notoriously hard to identify cell types at such granularity in PBMC single-cell datasets from healthy donors, we considered 9852 RNA-seq samples of purified cells from Ota et al. (2021) as references. Purified RNA-seq samples are average profiles for thousands of cells of the same cell type. To match cell types between the purified reference and flow cytometry from *Monaco bulk*, we harmonize cell type labels. Resulting ground truth and reference had 18 cell subsets defined are given in Appendix D.

## 4.2 EVALUATION METRICS

We used Pearson correlation and root mean squared error (RMSE) for the evaluation of deconvolution results. Since some cell types are much more abundant than others, it is important to consider the overall and per cell type average correlation and RMSE (see Apppendix B).

## 4.3 RESULTS

In this section, we present results on experiments detailed in Section 4.1. Additional experiments and results on datasets without corresponding flow cytometry fractions are presented in Appendix E. There we utilize validated biological hypotheses to evaluate DISSECT against competing methods.

### 4.3.1 ESTIMATION OF CELLTYPE FRACTIONS

To evaluate deconvolution performance, we deconvolved each of the datasets in Table 1 using the *PBMC8k* reference dataset (Table 2). For MuSiC, we also evaluate using all 10x PBMC datasets listed in Table 2 as well as blood data from Immune Cell Atlas (ICA) (Appendix table 7) since MuSiC can take advantage of multi-sample reference (3). Tables 3 and 4 demonstrate that DISSECT shows significantly improved correlations in 9 out of 12 comparisons and the lowest RMSE in 11 out of 12 comparisons, across 6 different datasets.

Next, we evaluated the cell fraction deconvolution performance on the *Monaco bulk* (Section 4.1) dataset that contains several closely related and rare cell types and constitutes a relatively harder task. With a correlation of 0.6, DISSECT's average performance is 14 percentage points better than the second best performance by Scaden (Appendix Table 8). For 8 out of 18 cell types it reaches the best correlation, while Scaden performs best for 3 out of 18 cell types. With an RMSE of 0.03, Scaden's performance is 1 percentage point better than DISSECT (Appendix Table 9). In summary, DISSECT displays the best correlation and a highly competitive RMSE in the cell fraction deconvolution task.

---

[1]https://support.10xgenomics.com/single-cell-gene-expression/datasets

Table 3: *Average (overall)* Pearson correlation between estimations and flow cytometry cell type fractions computed over all cell types.

| Dataset | MuSiC | MuSiC (All) | MuSiC (ICA) | CSx | Scaden | TAPE-O | TAPE-A | Linear MLPs | DISSECT |
|---|---|---|---|---|---|---|---|---|---|
| SDY67 | nan (-0.301) | 0.39 (-0.201) | 0.557 (0.23) | 0.59 (0.52) | 0.507 (0.419) | 0.53 (0.465) | 0.513 (0.152) | 0.51 (0.739) | **0.631** (**0.789**) |
| Monaco bulk | nan (0.006) | 0.534 (-0.4) | 0.684 (0.475) | 0.429 (0.634) | 0.685 (0.649) | 0.695 (0.569) | 0.664 (0.662) | 0.534 (0.73) | **0.727** (**0.783**) |
| Monaco Microarray | nan (-0.16) | nan (-0.418) | 0.592 (-0.41) | 0.66 (0.398) | **0.791** (0.189) | 0.71 (0.317) | nan (0.03) | 0.71 (0.233) | 0.786 (**0.799**) |
| GSE65133 | nan (-0.245) | nan (-0.3) | 0.554 ($-0.17$) | 0.642 (0.686) | 0.771 (0.413) | 0.729 (0.413) | 0.719 (0.252) | 0.77 (0.421) | **0.821** (**0.918**) |
| GSE107572 | nan (0.155) | 0.65 (0.24) | 0.542 (0.76) | 0.669 (0.527) | 0.627 (0.723) | 0.615 (0.658) | 0.012 (0.617) | 0.486 (0.71) | **0.705** (**0.79**) |
| GSE120502 | 0.103 (-0.302) | 0.451 (0.718) | 0.567 (0.77) | 0.529 (0.776) | 0.688 (0.718) | **0.692** (**0.883**) | 0.632 (0.779) | 0.63 (0.379) | 0.662 (0.864) |

Table 4: *Average (overall)* RMSE between estimations and flow cytometry cell type fractions computed over all cell types.

| Dataset | MuSiC | MuSiC (All) | MuSiC (ICA) | CSx | Scaden | TAPE-O | TAPE-A | Linear MLPs | DISSECT |
|---|---|---|---|---|---|---|---|---|---|
| SDY67 | 0.3 (0.348) | 0.201 (0.226) | 0.173 (0.21) | 0.147 (0.15) | 0.134 (0.13) | 0.11 (**0.10**) | 0.136 (0.12) | 0.13 (0.128) | **0.089** (**0.10**) |
| Monaco bulk | 0.247 (0.256) | 0.185 (0.194) | 0.113 (0.132) | 0.125 (0.094) | 0.081 (0.07) | 0.086 (0.097) | 0.065 (**0.068**) | 0.08 (0.077) | **0.06** (0.069) |
| Monaco Microarray | 0.27 (0.296) | 0.292 (0.304) | 0.21 (0.23) | 0.151 (0.134) | 0.139 (0.162) | 0.15 (0.15) | 0.17 (0.183) | 0.285 (0.307) | **0.07** (**0.069**) |
| GSE65133 | 0.308 (0.381) | 0.283 (0.292) | 0.2 (0.212) | 0.11 (0.121) | 0.13 (0.149) | 0.122 (0.119) | 0.14 (0.136) | 0.21 (0.253) | **0.06** (**0.058**) |
| GSE107572 | 0.245 (0.267) | 0.167 (0.196) | 0.099 (0.109) | 0.12 (0.137) | 0.105 (0.128) | 0.086 (0.099) | 0.116 (0.128) | 0.11 (0.11) | **0.068** (**0.076**) |
| GSE120502 | 0.358 (0.458) | 0.162 (0.172) | 0.101 (0.105) | 0.12 (0.12) | 0.104 (0.121) | 0.087 (0.093) | 0.093 (0.102) | 0.143 (0.15) | **0.086** (**0.087**) |

Finally, we performed an ablation study that validates our hypothesis that the consistency loss is primarily responsible for DISSECT's improved deconvolution performance (Appendix Table 18). We also evaluated the output of DISSECT at the end of simulation-phase only. These results are provided in Appendix Tables 17 (Correlation) and 18 (RMSE) where simulation-based phase lags behind the full consistency-regularized training.

### 4.3.2 ESTIMATION OF CELL TYPE-SPECIFIC GENE EXPRESSION

Next, we evaluated the performance of DISSECT's cell type-specific gene expression inference on simulated bulk RNA-seq data. We used simulated data as we could not obtain bulk RNA-seq and corresponding cell type-specific expression information. To maintain a domain shift between the training and test datasets, simulated data for training and testing were created using different single-cell datasets. Here, we compare our approach with the only two state-of-the-art methods that can infer cell type-specific gene expression per sample, TAPE and CSx (Section 3). We simulated bulk samples from one of the four reference single-cell PBMC datasets listed in Table 2, and created training simulations from the remaining three. To assess the performances, we computed sample-(Table 13) and gene-wise (Table 6) Pearson correlations. DISSECT displays the best sample- and gene-wise correlations in 3 out of 4 experiments, each, taking first place in overall cell type-specific gene expression deconvolution performance.

Table 5: Pearson correlation between ground truth and estimated gene expression profiles on simulated datasets averaged over samples. The column *Dataset* indicates the single-cell dataset used to create simulations for the test set.

| Dataset | CSx | TAPE-A | DISSECT |
|---------|-----|--------|---------|
| PBMC6k | $0.71 \pm 0.12$ | $\mathbf{0.83 \pm 0.09}$ | $0.82 \pm 0.08$ |
| PBMC8k | $0.82 \pm 0.13$ | $0.79 \pm 0.09$ | $\mathbf{0.84 \pm 0.11}$ |
| DonorA | $0.78 \pm 0.12$ | $0.85 \pm 0.11$ | $\mathbf{0.89 \pm 0.10}$ |
| DonorC | $0.75 \pm 0.09$ | $0.81 \pm 0.12$ | $\mathbf{0.83 \pm 0.08}$ |

Table 6: Pearson correlation between ground truth and estimated gene expression profiles on simulated datasets averaged over estimated genes.

| Dataset | CSx | TAPE-A | DISSECT |
|---------|-----|--------|---------|
| PBMC6k | $0.37 \pm 0.22$ | $0.42 \pm 0.14$ | $\mathbf{0.46 \pm 0.15}$ |
| PBMC8k | $0.36 \pm 0.25$ | $\mathbf{0.51 \pm 0.12}$ | $0.48 \pm 0.14$ |
| DonorA | $0.42 \pm 0.21$ | $0.48 \pm 0.20$ | $\mathbf{0.48 \pm 0.18}$ |
| DonorC | $0.46 \pm 0.18$ | $0.45 \pm 0.11$ | $\mathbf{0.49 \pm 0.12}$ |

## 5 DISCUSSION

We detailed how the use of a linear consistency is suitable for the task of deconvolution, especially in the absence of real ground truth training information, as is often the case in biomedical settings. Our approach relies on the supervised learning on simulated data and an unsupervised domain adaptation to the target data of interest. This semi-supervised learning approach results in state-of-the-art deconvolution performance, for both cell fraction and gene expression estimation. While we only focused on MLPs for estimation of cell type fractions and autoencoders for gene expression estimation in this work, we surmise that consistency regularization might improve other deconvolution algorithms as well. We envision further work in this area.

The task of deconvolution plays an important role in spatial transcriptomics (ST) and cell-free DNA methylation (cfDNA). Recently, several algorithms have been developed for ST (Li et al., 2022) and cfDNA deconvolution (Jeong et al., 2022). We surmise that consistency regularization might also improve ST and cfDNA deconvolution by adjusting DISSECT's simulation procedure to mimic ST or cfDNA. We provide two proof-of-concept ST deconvolution results using consistency regularization in Appendix F. Similar arguments can be made for the deconvolution of several other biomedical data types, such as epigenetic, proteomic, and metabolomic data, for instance.

## 6 LIMITATIONS

While DISSECT displays favorable deconvolution performance compared to other methods, the results are far from perfect. Especially for hard deconvolution tasks, such as samples with many similar cell types and very scarce cell populations, an increase in deconvolution performance is warranted. Future research into semi-supervised and contrastive algorithms as well as data augmentation and integration techniques should further enhance DISSECT's performance. As stated in Section 4.3.2 we had to rely on the simulation based experiment to evaluate gene expression estimations. Nevertheless, we still explored how well DISSECT can estimate gene expression using an ST dataset (Appendix H). Further evaluations with quality ground truths will be beneficial.

## 7 CODE AND DATA AVAILABILITY

All considered datasets are publicly available from respective sources. Code is available anonymously at `https://anonymous.4open.science/r/DISSECT-F0C4`.

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

A

Table 7: Details on Blood single-cell data from Immune Cell Atlas (Domínguez Conde et al. (2022)) used as a multi-sample reference scRNA-seq for MuSiC. Cell types that were present in less than 5 samples were dropped. The same cell types were selected as in scRNA-seq datasets listed in Table 2: Bcells, CD4Tcells, CD8Tcells, Monocytes and NK. The rest of the cell types were merged to form the unknown cluster (Section 4.1).

| Donor name | # cells | Post-QC number of genes |
|---|---|---|
| 621B | 103 | 23658 |
| 637C | 760 | 23658 |
| A35 | 1368 | 23658 |
| A36 | 3124 | 23658 |
| D496 | 9065 | 23658 |
| D503 | 12208 | 23658 |

Table 8: Pearson correlation between estimates from different methods and flow cytometry for granular cell type fractions in *Monaco bulk*.

| Celltype | MuSiC | CSx | Scaden | TAPE-O | TAPE-A | Linear MLPs | DISSECT |
|---|---|---|---|---|---|---|---|
| B Ex | nan | **0.43** | 0.18 | 0.22 | 0.030 | 0.10 | 0.32 |
| B NSM | nan | -0.08 | **0.12** | 0.10 | -0.15 | -0.22 | 0.09 |
| B Naive | nan | 0.95 | 0.87 | 0.8 | 0.43 | 0.71 | **0.96** |
| B SM | **0.85** | nan | 0.57 | 0.45 | 0.15 | 0.26 | 0.63 |
| Monocytes C | 0.30 | 0.29 | **0.63** | 0.57 | 0.52 | 0.11 | 0.62 |
| Monocytes I | 0.41 | 0.36 | 0.90 | 0.87 | 0.81 | 0.54 | **0.93** |
| Monocytes NC | 0.25 | 0.09 | 0.31 | 0.35 | 0.48 | 0.19 | **0.66** |
| NK | 0.80 | **0.82** | 0.58 | 0.59 | 0.65 | 0.49 | **0.82** |
| Neutrophils LD | 0.2 | nan | **0.89** | 0.48 | 0.57 | 0.03 | 0.56 |
| Plasmablasts | 0.62 | 0.85 | 0.86 | 0.65 | 0.66 | 0.42 | **0.92** |
| CD4 T Naive | 0.66 | 0.47 | 0.68 | 0.70 | 0.34 | 0.14 | **0.76** |
| CD4 T Memory | **0.47** | -0.15 | 0.27 | 0.27 | 0.12 | 0.08 | 0.24 |
| CD8 T Naive | 0.52 | **0.7** | 0.36 | 0.38 | 0.32 | 0.27 | 0.49 |
| CD8 T CM | nan | -0.65 | 0.19 | 0.13 | **0.21** | 0.01 | 0.12 |
| CD8 T EM | nan | nan | 0.02 | 0.47 | 0.45 | 0.11 | **0.62** |
| CD8 T TE | 0.25 | **0.9** | 0.28 | 0.35 | 0.36 | 0.35 | 0.86 |
| mDC | nan | 0.46 | 0.47 | 0.39 | 0.40 | 0.05 | **0.68** |
| pDC | 0.55 | **0.57** | 0.19 | 0.42 | 0.31 | 0.3 | 0.55 |
| **Average** | 0.49 [2] | 0.40 [2] | 0.46 | 0.45 | 0.37 | 0.22 | **0.60** |

B NOTE ON AVERAGE AND OVERALL CORRELATIONS FOR PERFORMANCE EVALUATION.

Overall metrics can be deceiving when the model performs comparatively very well or very poor on a largely abundant cell type as illustrated by results in Figures 2 and 3. Overall metrics are provided for completeness as it has been extensively used in evaluation of other deconvolution works and benchmarks (Avila Cobos et al., 2020; Newman et al., 2019; Wang et al., 2019; Chen et al., 2021).

C QUALITY CONTROL AND PREPROCESSING

C.1 QUALITY CONTROL:

Before simulating from reference datasets, we remove cells with less than 200 expressed genes and genes which are expressed in less than 3 cells. Further, we also remove cells expressing more than 4% mitochondrial genes. Thereafter, before each deconvolution, we subset reference and bulk

---

[2]Calculated over real values.

Table 9: RMSE between estimates from different methods and flow cytometry for granular cell type fractions in *Monaco bulk*.

| Celltype | MuSiC | CSx | Scaden | TAPE-O | TAPE-A | Linear MLPs | DISSECT |
|----------|-------|-----|--------|--------|--------|-------------|---------|
| B Ex | **0.01** | 0.05 | 0.04 | 0.04 | **0.01** | 0.02 | 0.02 |
| B NSM | 0.02 | **0.01** | 0.02 | 0.03 | 0.05 | 0.04 | 0.02 |
| B Naive | **0.01** | 0.03 | 0.03 | 0.03 | 0.04 | 0.06 | 0.03 |
| B SM | 0.05 | **0.01** | **0.01** | 0.02 | 0.02 | 0.03 | **0.01** |
| Monocytes C | 0.05 | **0.02** | 0.04 | **0.02** | **0.02** | 0.06 | 0.06 |
| Monocytes I | 0.12 | 0.15 | **0.03** | 0.06 | 0.10 | 0.12 | 0.04 |
| Monocytes NC | 0.20 | 0.09 | **0.02** | 0.07 | 0.05 | 0.10 | **0.02** |
| NK | **0.05** | 0.08 | 0.08 | 0.11 | 0.11 | 0.05 | 0.08 |
| Neutrophils LD | 0.02 | 0.03 | **0.01** | **0.01** | **0.01** | 0.01 | 0.02 |
| Plasmablasts | **0.01** | **0.01** | 0.02 | **0.01** | 0.04 | 0.01 | 0.04 |
| CD4 T Naive | **0.02** | 0.03 | 0.05 | 0.05 | **0.02** | 0.02 | 0.05 |
| CD4 T Memory | 0.10 | 0.07 | **0.03** | 0.12 | 0.15 | 0.21 | **0.03** |
| CD8 T Naive | 0.21 | 0.07 | **0.04** | 0.05 | 0.05 | 0.01 | **0.04** |
| CD8 T CM | **0.01** | 0.08 | 0.02 | 0.05 | 0.12 | 0.01 | 0.03 |
| CD8 T EM | 0.02 | **0.01** | 0.02 | 0.02 | **0.01** | 0.03 | 0.02 |
| CD8 T TE | **0.01** | 0.07 | 0.09 | 0.08 | 0.11 | 0.16 | 0.08 |
| mDC | **0.01** | 0.04 | 0.04 | 0.08 | 0.09 | 0.03 | 0.03 |
| pDC | 0.02 | **0.00** | 0.02 | 0.01 | 0.01 | 0.01 | 0.02 |
| **Average** | 0.06 | 0.05 | **0.03** | 0.05 | 0.06 | 0.05 | 0.04 |

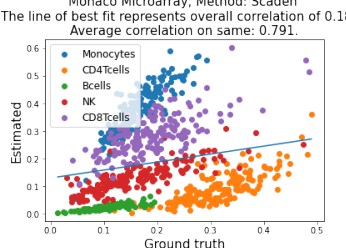

Figure 2: Correlations between Ground truth fractions from *Monaco Microarray* (Table 1) and the estimations using Scaden.

Figure 3: Correlations between the ground truth fractions from *GSE120502* (Table 1) and the estimations using TAPE-O.

datasets to include only the common genes between the two. This quality control step was identical for all methods.

## C.2 SIMULATIONS

For deep learning methods, we sampled $\alpha_{k,i}$ uniformly to simulate based on procedure described in Section 2 with $\sum_{k=1}^{c} \alpha_{k,i} = 100, \forall i$ if the dataset is single-cell. For experiments on granular level cell types where simulations are done from purified cell samples, we modify simulation procedure to reflect this. In this case, a simulated sample is given by $\mathbf{B}_{i\cdot}^{\mathrm{sim}} = \sum_{k=1}^{c} \mathbf{X}_{ik}^{\mathrm{sim}} \mathbf{b}_{\mathbf{l}}^{\mathbf{k}}$, where $\mathbf{b}_{\mathbf{l}}^{\mathbf{k}}$ is the expression vector of purified sample $l$ belonging to cell type $k$. All other notations are same as in Section 2.

For all experiments, we simulated total $1000 \times c$ simulations where $c$ is number of cell types in the reference dataset.

## C.3 Preprocessing:

**Estimation of cell type fractions:**

**Scaden, TAPE, Linear MLPs and DISSECT:** Before passing simulated and real bulk samples to the network, we normalize samples to sum to a million counts (CPM: Counts per million) and log scale them with base 2 after adding 1. CPM normalization was performed to maintain total mRNA expressed per gene to be out of a fixed total gene expression, and CPM is widely used in computational genomics. During training, for each batch, we normalize each sample by $MinMax$ scaling. These are standard preprocessing steps (Menden et al., 2020).

For MuSiC and CibersortX (under S-mode), data was supplied on a linear scale as suggested in their respective publications and no change was made to the default normalization methods of both (Wang et al., 2019; Newman et al., 2019).

**Estimation of per sample cell type specific gene expression profiles:** To estimate cell type specific gene expression profiles, we need to maintain relationship between gene expression of individual cell types and simulated bulks, which would be lost if we perform CPM normalization of both simulated samples and corresponding cell type specific gene expression profiles. Hence, instead of performing CPM normalization of simulated bulks, we normalize each test bulk sample to sum to the mean of sums of simulated samples. Further, for estimating cell type specific gene expression, we want to maintain gene level information across samples. To achieve this, instead of normalizing each sample using $MinMax$ scaling, we perform $MinMax$ scaling globally over all samples.

For TAPE, since the signature matrix is observed in decoder (Section 3), preprocessing step is similar to the preprocessing done in estimating cell type fractions. For CibersortX, data was supplied on a linear scale under S-mode (Newman et al., 2019).

## D Cell subsets for estimations of granular subsets on Monaco bulk.

**B cells (4 subsets):** B Naive (Naive B cells), B Ex (Exhausted B cells), B NSM (Non-switched memory B cells), B SM (Switched memory B cells)

**CD4 T cells (2 subsets):** CD4 T Naive (Naive CD4 T cells), CD4 T Memory (Memory CD4 T cells),

**CD8 T cells (4 subsets):** T CD8 Naive (Naive CD8 T cells), CD8 T CM (Central Memory CD8 T cells), CD8 T TE (Terminally effector CD8 T cells), and CD8 T EM (Effector Memory CD8 T cells),

**Monocytes (3 subsets):** Monocytes C (Classical monocytes), Monocyte NC (Non classical monocytes) and Monocytes I (Intermediate monocytes),

**Dendritic cells (2 subsets):** mDC (myeloid dendritic cells), pDCs (Plasmacytoid dendritic cells)

**Plasmablasts**

**Neutrophils LD** (Low density neutrophils)

**NK cells**.

## E Further evaluation of deconvolution performance using diverse tissue datasets.

To assess the performance of DISSECT and other algorithms on further bulk RNA-seq datasets, we consider additional experiments. In Section E.1, we consider paired scRNA-seq and bulk RNA-seq data. In Section E.2, we looked at the performance of the methods to recover established biological findings and in Section E.3, we assessed how the performance changes when the reference scRNA-seq dataset is swapped with another reference.

Table 10: Harmonization of cell subset labels between *Monaco bulk* and reference from Ota et al. (2021)

| Cell subset | Subset(s) in flow cytometry of *Monaco bulk* | Merged subset(s) in Ota et al. (2021) |
|---|---|---|
| B Ex | B Ex | DN_B |
| B NSM | B NSM | USM_B |
| B Naive | B Naive | Naive_B |
| B SM | B SM | SM_B |
| Monocytes C | Monocytes C | CL_Mono |
| Monocytes I | Monocytes I | Int_Mono |
| Monocytes NC | Monocytes NC | NC_Mono |
| NK | NK | NK |
| Neutrophils LD | Neutrophils LD | LDG |
| Plasmablasts | Plasmablasts | Plasmablast |
| CD4 T Naive | T CD4 Naive | Naive_CD4 |
| CD4 T Memory | Th1, Th2,Th17,Th1/Th17, Tregs | Th1, Th2, Th17, Tfh, Fe_II_eTreg |
| CD8 T Naive | T CD8 Naive | Naive_CD8 |
| CD8 T CM | T CD8 CM | CM_CD8 |
| CD8 T EM | T CD8 EM | EM_CD8 |
| CD8 T TE | T CD8 TE | TEMRA_CD8, Mem_CD8 |
| mDC | mDCs | mDC |
| pDC | pDCs | pDC |

### E.1 PAIRED SCRNA-SEQ AND BULK RNA-SEQ

To evaluate deconvolution methods on further bulk RNA-seq datasets, we obtained paired scRNA-seq and bulk RNA-seq from two tissues: mammary gland and lung. The details of these datasets are provided in Table 11. The ground truth for bulk RNA-seq was generated using the fractions of cell types as observed in the scRNA-seq.

The Tables 12 and 13 present the results on these two tissues. For the mammary gland dataset, the results are calculated per sample and averaged since the dataset contains only two samples as done in the original publication of data.

Table 11: Details on paired single-cell and bulk RNA-seq datasets considered. Cell type labels were used as provided in the corresponding sources.

| Dataset name | # Post-QC cells | # samples (bulk) | Original Source |
|---|---|---|---|
| Mammary gland | 3991 | 2 | Dong et al. (2021) |
| Lung | 93246 | 17 | Delorey et al. (2021) |

Table 12: Average Pearson correlation and RMSE between ground truth and estimated cell type fractions on the *Lung* dataset over all cell types.

| Dataset | MuSiC | CSx | Scaden | TAPE-O | TAPE-A | Linear MLPs | DISSECT |
|---|---|---|---|---|---|---|---|
| r | 0.51 | 0.56 | 0.55 | 0.53 | 0.54 | 0.48 | 0.56 |
| rmse | 0.09 | 0.07 | 0.07 | 0.08 | 0.08 | 0.12 | 0.06 |

### E.2 RELATIONSHIP BETWEEN CELL TYPE FRACTIONS AND BIOLOGICAL PHENOTYPES

In this section, we rely on the established biological findings to evaluate deconvolution methods. For this purpose we considered diverse set of tissues: brain, kidney and pancreas. Table 14 lists these datasets and corresponding hypothesis based on literature. The single cell datasets corresponding to these tissues are presented in 15.

Here we are interested in investigating if the deconvolution methods are faithful to the established biological findings (Presented and discussed further in this Section). We are also interested in how

Table 13: Average Pearson correlation and RMSE between ground truth and estimated cell type fractions on the *Mammary gland* dataset over samples.

| Dataset | MuSiC | CSx | Scaden | TAPE-O | TAPE-A | Linear MLPs | DISSECT |
|---------|-------|------|--------|--------|--------|-------------|---------|
| r | 0.85 | 0.87 | 0.77 | 0.84 | 0.81 | 0.74 | 0.91 |
| rmse | 0.09 | 0.07 | 0.1 | 0.09 | 0.1 | 0.1 | 0.06 |

Table 14: Details on bulk RNA-seq datasets used to evaluate deconvolution methods on biological phenotypes. Biological hypotheses based on literature serve as proxy ground truths.

| Tissue | Dataset | # samples | Biological hypothesis based on literature | original Source |
|--------|---------|-----------|-------------------------------------------|-----------------|
| Pancreas | GSE50244 | 89 (77 with information on hemoglobic 1C levels) | Fraction of beta cells are negatively associated with severity of type 2 diabetes indicated by hemoglobin A1c (hba1C) level (Alejandro et al., 2015; Saisho, 2015; Wang et al., 2013). | Fadista et al. (2014) |
| Kidney | GSE81492 | 10 | Tubule cells diminish with chronic kidney disease (CKD) (Venkatachalam et al., 2015; Liu et al., 2018; Malhotra et al., 2019). | Beckerman et al. (2017) |
| Brain | ROSMAP | 508 (463 with corresponding annotation of Braak stages) | 1. Neurodegeneration with advanced Braak stage (Streit et al., 2009; Hindle, 2010; Fu et al., 2019), and 2. Approximately 70-30 ratio of excitatory and inhibitory neurons (Contreras, 2004; Chen & Dzakpasu, 2010). | Bennett et al. (2018) |

methods behave when different single-cell reference datasets are used (Presented and discussed in Section E.3).

**Deconvolution of *ROSMAP* with reference scRNA-seq from *Allen Brain Atlas*:**

*ROSMAP* cohort consists of samples from healthy individuals and patients with Alzheimer's disease (AD). Here, we consider two biological ground truths: first is the neurodegeneration, or the loss of neurons with increasing Braak Stages (Braak et al., 2003) (Table 14), and the second is the ratio of excitatory neurons to inhibitory neurons. We deconvolved the *ROSMAP* using reference from *Allen Brain Atlas*. The results are presented in Figure 8. Nearly all methods capture a negative association between the median fractions of excitatory neurons and Braak stages. Scaden, TAPE-O, DISSECT maintain a higher proportion of excitatory neurons compared to inhibitory neurons. However, DISSECT estimates show the excitatory-inhibitory neurons ratio to be almost 70-30. TAPE-A and MuSiC on the other hand show opposite of what is expected.

**Deconvolution of *GSE50244* with reference scRNA-seq from *Segerstolpe*:**

*GSE50244* is a bulk RNAseq dataset from pancreas and consists of samples from healthy and T2D (Type 2 diabetes) individuals. Here our biological ground truth is the negative association between beta cell proportions and the level of hemoglobin A1c (hba1C) (Table 14). We performed the deconvolution using *Segerstolpe*. We restricted dataset to contain alpha, beta, gamma, delta, acinar and ductal cell types following the methodology in Wang et al. (2019). The results are presented in Figure 4. W note that all deconvolution methods successfully reveal the significant association between beta cell proportions and hba1c level, however since the extent of the association is unknown, further quantification would be speculative.

**Deconvolution of *GSE81492* with reference scRNA-seq from *Park*:**

Table 15: Details on single-cell datasets used to deconvolve corresponding tissue samples.

| Tissue | Dataset name | # cells | Original Source |
|---|---|---|---|
| Pancreas | Baron | 8569 | Baron et al. (2016) |
| Pancreas | Segerstolpe | 3514 | Segerstolpe et al. (2016) |
| Pancreas | Xin | 1492 | Xin et al. (2016) |
| Kidney | Park | 43745 | Park et al. (2018) |
| Kidney | Miao | 16887 | Miao et al. (2021) |
| Brain | Allen Brain Atlas | 49418 | bra. |

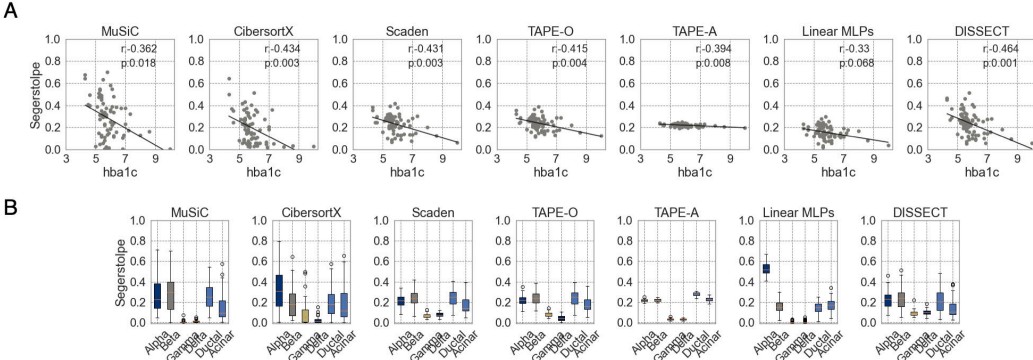

Figure 4: **A:** Distribution of relative fractions of alpha, beta, gamma, delta, acinar and ductal cells estimated on 77 bulk RNA-seq samples from *GSE52044* using *Seger* single-cell reference dataset. B: Relationship between Hemoglobin A1C (hba1c) levels and estimated fractions of beta cells in 77 bulk RNA-seq samples from *GSE52044* using *Seger*. Corresponding Method for each plot is indicated in the title. Pearson correlation of the relationship and associated *p*-value is indicated in the plot. *p*-value shown is obtained for beta cells from a multiple linear regression model considering age, sex and bod mass index (BMI) as covariates, *i.e.* model hb1ac $\sim$ Constant + fractions of beta cells + Age + Sex + BMI.

*GSE81492* is a dataset consisting of APOL1 mutant mice (a Chronic Kidney, CKD, disease mouse model) (Table 14). Here the biological ground truth is the decrease in tubule cells - Proximal tubule (PT) cells ductal convoluted tubule (DCT) cells in CKD samples compared to the healthy state. We deconvolved the aforementioned dataset using single cell reference dataset *Park*. We present the results per method per reference in Figure 5. All methods reveal loss of proximal tubule (PT) cells in APOL1 mice, while showing higher proportion of immune cells (B lymph, Fib, Maco and NK cells) in APOL1. However, DCT cells (Ductal convoluted tubule) cells which are also known to diminish in the APOL1 mice, MuSiC shows an increase. This is also observed in Wang et al. (2019). However, other methods were successful in revealing loss of DCTs in APOL1.

### E.3 RELATIONSHIP BETWEEN CELL TYPE FRACTIONS AND BIOLOGICAL PHENOTYPES USING MULTIPLE REFERENCE SCRNA-SEQ

**Deconvolution of *GSE50244* with reference scRNA-seq from *Segerstolpe, Baron* and *Xin*:**

For *GSE50244*, we performed deconvolution using three reference single-cell datasets. These references differ in technologies with which the cells were sequenced. These technologies are *Baron*: inDrop, *Segerstolpe*: Smart-seq2, and *Xin*: SMARTer. Further, these reference datasets contain cells belonging to different states. We used cells belonging to only healthy individuals in *Baron* and *Segerstolpe* while both healthy and T2D individuals are used in *Xin*. Further, since our goal is to compare the divergence in performance when the reference single-cell dataset is changed, we subsetted all three single-cell datasets to contain same cell types. Figure 6 shows the distribution of alpha, beta, gamma and delta cells for each deconvolution method. There is a lack of concordance between three distributions of cell types across all methods. Beta and alpha cells are generally the

two most abundant of these four cell types in pancreatic islets (Henquin & Rahier, 2011). This is correctly observed with estimations from the considered deep learning based methods. However, for Scaden, the relative proportions of alpha and beta cells are inverted between *Baron* and *Xin*. While for DISSECT, they are predicted almost at the same level for the three datasets, which more varying beta cell fractions when *Baron* is the reference. TAPE-O also achieves this trend, however, TAPE-O incorrectly predicts Delta cells as being at the same level as alpha cells for reference *Baron*. Linear MLPs show the most variance and predict almost 80% alpha cells for *Segerstolpe* and *Xin*. Further, with Linear MLPs, the least abundant gamma and delta cells are predicted to be negative for almost all samples. Next, we looked at the association between beta cell fractions and T2D severity (Table 14). Across all three datasets, DISSECT estimations are significantly negatively correlated with hb1ac. As in other experiments, we observe a wide discrepancy between TAPE-O and TAPE-A results.

**Deconvolution of *GSE81492* with reference scRNA-seq from *Park* and *Miao*:**

We deconvolved *GSE81492* using two single cell reference datasets *Park* and *Miao*. Here, we subsetted these two reference datasets to contain same cell types. We present the results per method per reference in Figure 9. To enable comparisons, same *y*-axes were used for both single-cell datasets. Both of these datasets come from same technologies (10x Genomics). Despite this, almost all methods show variation in their estimates when changing the reference, wit DISSECT showing the least variance and giving similar associations between cell type fractions and tissue state.

These experiments show that while deconvolution methods in general follow biology but the results differ across the single-cell reference used. DISSECT, however, shows more robustness compared to other methods in this regard.

## F    APPLICATION TO SPATIAL TRANSCRIPTOMICS

Here we illustrate applicability of DISSECT on spatial transcriptomics (ST). We focused on two publicly available tissue slides (Mouse brain and human lymph node) on which ST has been performed.

In brain, cortical neuronal layers are structured spatially. To verify whether DISSECT estimates may be valid in ST, we deconvolved a sagittal mouse brain slice available as part of Seurat [3]. As reference, we used a mouse brain single-cell dataset from Allen Brain Institute consisting of approximately 14000 cells sequenced using Smart-seqv2 protocol (Tasic et al., 2016). We adjusted simulation procedure to mimic ST datasets. 10x Visium (one of the technologies to generate ST samples) consists of around 10 cells per spot [4] [5].To reflect this, we simulated between 5-12 cells to generate one spot (*i.e.* $\sum_{k=1}^{c} \alpha_{ki} \sim [5, 12]$). Since ST is much sparser, to generate one spot, we kept between 2-6 cell types. Figure 10 shows fractions of cell types overlaid on the hematoxylin and eosin (H&E) stained images of tissue slide, and Figure 11 shows jointly cortical neuronal proportions which shows a spatially structured arrangement of neurons.

To evaluate how DISSECT behaves on ST deconvolution on granular level subsets, we deconvolved a human lymph node slide using corresponding integrated single-cell datasets on which granular level cell types are annotated. Both of these datasets are obtained from (Kleshchevnikov et al., 2022). Remarkably, DISSECT is able to identify spatial patterns of cell type fractions, along with correctly predicting co-localization of cycling B cells and germinal center B cells. This is illustrated in Figures 13 and 14. Several germinal center zones are correspondigly visible in h&E stained image (Figure 13).

These results demonstrate the usability of DISSECT on spatial deconvolution and warrant evaluation of consistency loss further in data modalities other than bulk RNA-seq.

---

[3] https://satijalab.org/seurat/articles/spatial_vignette.html

[4] Each spot is a location which is sequenced in tissue slide. Thus, each spot is analogous to a bulk RNA-seq, albeit much sparser due to less number of cells per spot.

[5] https://kb.10xgenomics.com/hc/en-us/articles/360035487952-How-many-cells-are-captured-in-a-single-spot-

## G    ABLATION

Table 16: Average performance over five random experiments for SDY67 (Table 1.) Each column indicates the additional part.

| Metric | Linear MLP | Activations | KL Divergence | KL Divergence + Consistency |
|---|---|---|---|---|
| r | $0.51 \pm 0.018$ | $0.55 \pm 0.016$ | $0.54 \pm 0.006$ | $0.63 \pm 0.005$ |
| rmse | $0.13 \pm 0.008$ | $0.13 \pm 0.006$ | $0.11 \pm 0.004$ | $0.09 \pm 0.002$ |

Table 17: Average Pearson correlation between estimations and flow cytometry cell type fractions for the only after simulation phase (Step 2000, $\lambda = 0$) in comparison to the full training.

| Dataset | Simulation-phase only | DISSECT |
|---|---|---|
| SDY67 | 0.359 | 0.631 |
| Monaco bulk | 0.698 | 0.727 |
| Monaco microarray | 0.682 | 0.786 |
| GSE65133 | 0.66 | 0.821 |
| GSE107572 | 0.53 | 0.705 |
| GSE120502 | 0.64 | 0.662 |

Table 18: Average RMSE between estimations and flow cytometry cell type fractions for the only after simulation phase (Step 2000, $\lambda = 0$) in comparison to the full training.

| Dataset | Simulation-phase only | DISSECT |
|---|---|---|
| SDY67 | 0.226 | 0.089 |
| Monaco bulk | 0.081 | 0.06 |
| Monaco microarray | 0.139 | 0.07 |
| GSE65133 | 0.14 | 0.06 |
| GSE107572 | 0.131 | 0.068 |
| GSE120502 | 0.09 | 0.086 |

## H    ESTIMATING CELL TYPE SPECIFIC GENE EXPRESSION ON A BRAIN ST SLIDE

To evaluate DISSECT on the estimation of cell type specific gene expression, we utilize simulations from PBMC scRNA-seq datasets (Table 2). This is due to the unavailability of bulk RNA-seq from paired tissue and cell populations. However, to investigate further how DISSECT performs in practice, we investigated quallity of gene expression estimation for brain ST using scRNA-seq data from Allen Brain Atlas. Details of both datasets are provided in Section F. In this experiment, we were interested in answering two questions: 1. Does DISSECT estimates of gene expression reflect what is observed for that cell type in the literature? 2. Can DISSECT identify heterogeneity of the same cell type across samples (in this case spots) without having to pre-annotate cell type subsets? To accomplish answering of the second question, we merged excitatory neuronal subsets together and labeled them as "exc_neurons". This allows us to test whether we observe heterogeneity in excitatory neurons after estimation or not. This resulted in 17 final cell types. We also filtered our cells where the proportion of corresponding cell type is less than 1/(no. of cell types). This is reasonable as 10x Visium spots contain between 1-10 cells.

The first question relates to accuracy of the predicted signature, and the second question is about whether the biological reality of the sample at hand is preserved. To evaluate our results, we first computed PCA and UMAP embeddings of the predicted cell type specific gene expression and identified disjoint cell type clusters (Figure 15). Second, we tested for differential expression (DE) of genes for each cell type. Top 5 DE genes for each cell type are visualized over the ST slide (Figure 16). To verify whether the DE genes make sense in the broader context of the literature, we

performed gene set enrichment using Enrichr [6] with gene sets available from PanglaoDB Franzén et al. (2019), a curated database of single-cells from different tissues. The results are presented in Figures 17 and 18. Correct gene sets are enriched for each predicted signature (*e.g.* Astrocyte for cluster "Astro", Neurons or interneurons for neuronal subsets such as Lamp5, exc_neurons etc., Microglia and Macrophage for "Macrophage", Oligodendrocytes for the cluster "Oligo"). Next, we focused on the excitatory neurons. Figure 19 shows the expression of expected positive and negative marker genes (taken from the Allen Brain Atlas) over the ST slide. We observe that excitatory neurons do express positive markers but not the negative ones. This positively supports the first of our aforementioned questions.

To investigate whether we observe spatial heterogeneity in the predicted gene expression of excitatory neurons, we performed unsupervised clustering using louvain clustering with default resolution of 1. Figure 20 shows the louvain clusters over UMAP and over the ST slide. We observe that clusters have spatial variability and may be linked with the location. To verify this further, we looked at some genes, Cux2, Rorb and Fezf2, which are used in creating a neuronal taxonomy and in situ validation of excitatory cell types (Hodge et al. (2019), Tasic et al. (2016)). We observe the expression of these genes at the correct spatial locations (Cux2, Rorb and Fezf2 in this order with increased depth). Further, since these are only three genes and taken from the literature, we wanted to look into what the genes differentially expressed amongst these 9 clusters indicate. To this end, we performed DE analysis and used *Allen Brain Atlas Up* gene sets that are included in Enrichr [6]. Here we couldn't use PanglaoDB as it does not provide detailed taxonomy gene sets of neurons. Figure 21 presents the results of the gene set enrichment. In total, six of the clusters resulted in DE gene sets with default settings (p adjusted value cutoff of 0.01 and absolute log2 fold change cutoff of 1). We identify that each cluster is associated with certain brain regions. This positively supports our second question regarding identification of heterogeneity within a cell type label.

---

[6]https://maayanlab.cloud/Enrichr/

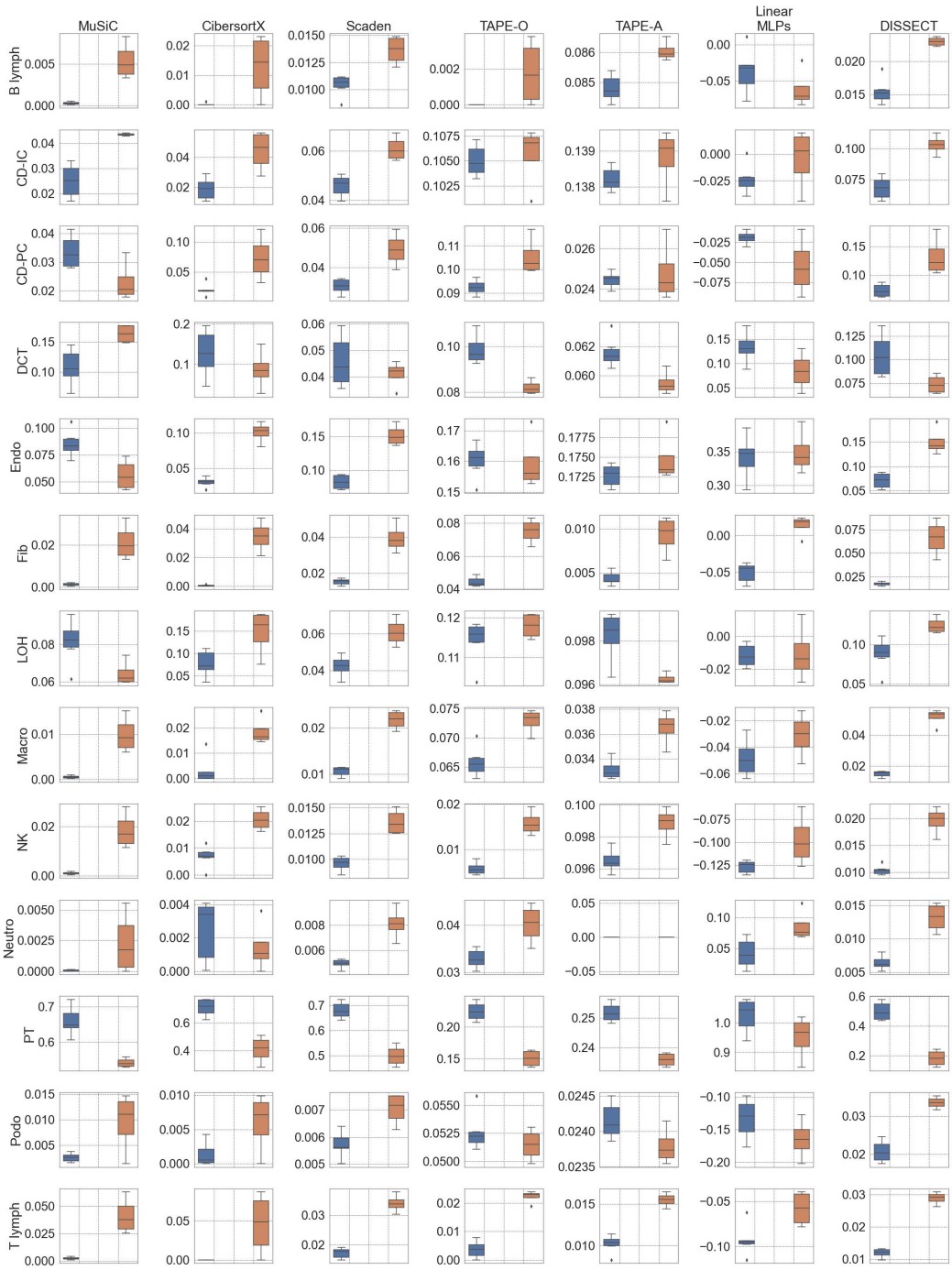

Figure 5: Estimated fractions of different cell types in 10 bulk RNA-seq samples from *GSE81492* (Ctrl: Control mice, n=6 and APOL1: Apolipoprotein L1 transgenic mice, n=4) mice using single-cell reference dataset *Park*. Corresponding Methods are indicated in the title. Each row corresponds to a cell type. DCT: Distal convoluted tubule, Endo: Endothelial cells, LOH: Loope of Henle, Macro: Macrophages, Neutro: Neutrophils, PT: Proximal Tubule, Podo: Podocytes, CD-PC: collecting duct principal cell; CD-IC: collecting duct intercalated cell.

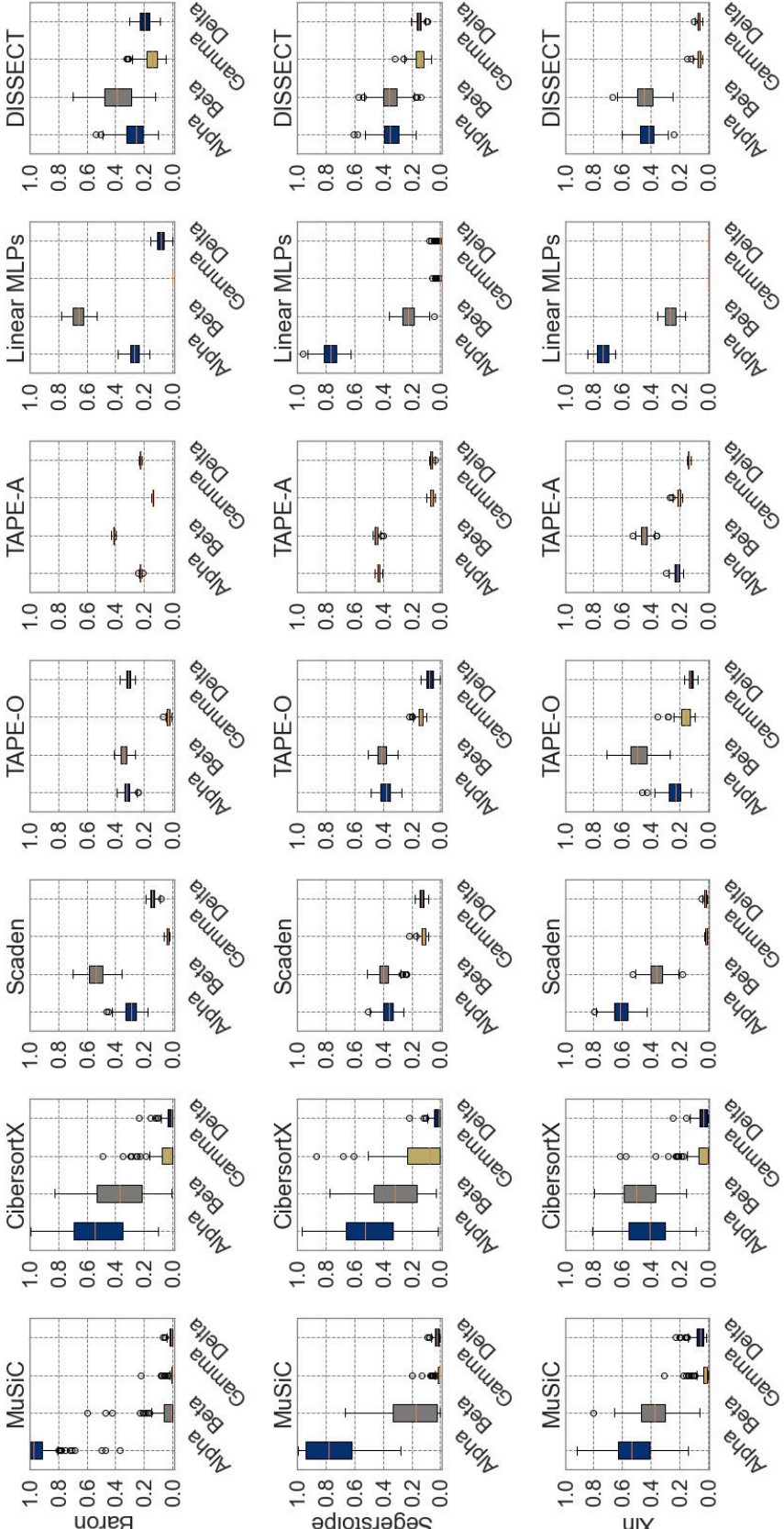

Figure 6: Distribution of relative fractions of alpha, beta, gamma and delta cells (present in all three single cell datasets) estimated on 77 bulk RNA-seq samples from *GSE52044* using three different single-cell reference datasets. Each row corresponds to the single-cell reference dataset.

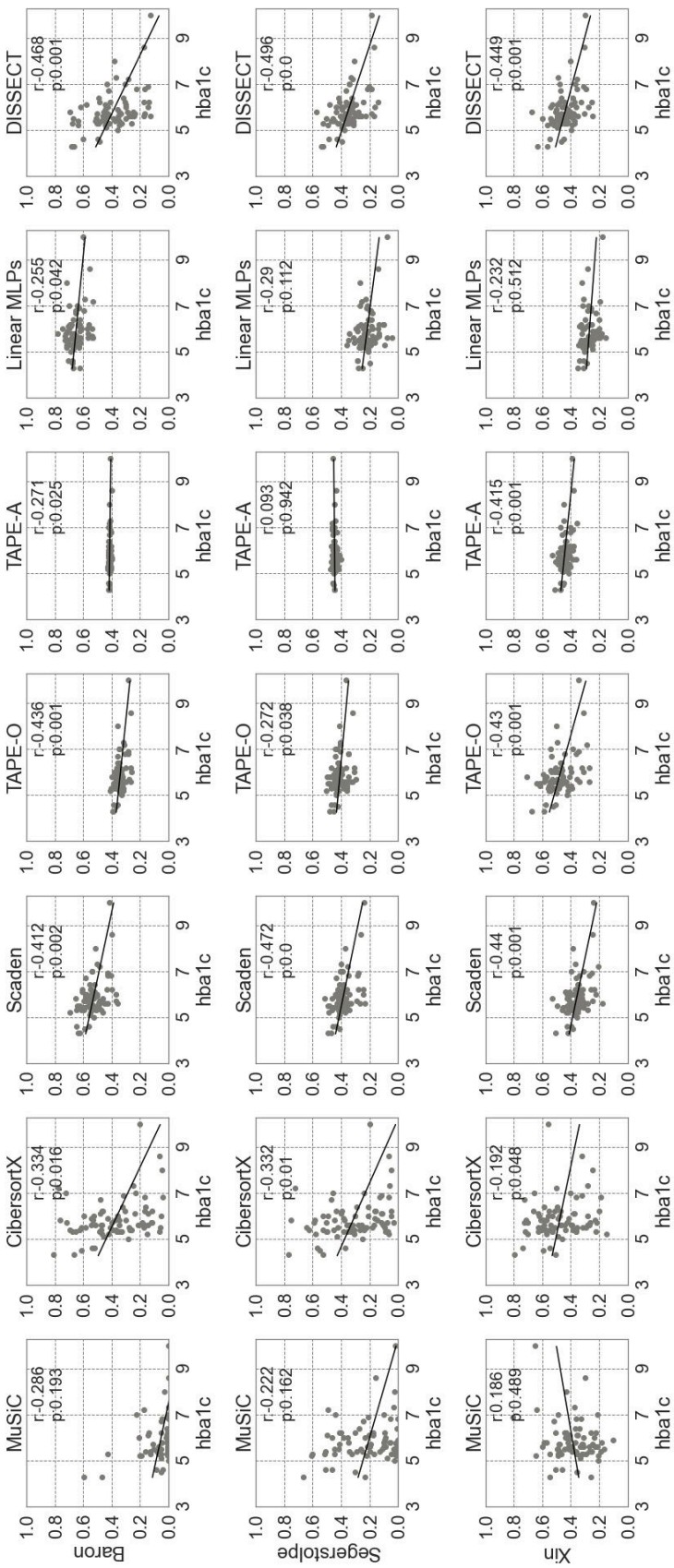

Figure 7: Relationship between Hemoglobin A1C (hba1c) levels and estimated fractions of beta cells in 77 bulk RNA-seq samples from *GSE52044* using three different single-cell reference datasets. Corresponding Method for each plot is indicated in the title. Each row corresponds to the single-cell reference dataset. Pearson correlation of the relationship and associated *p*-value is indicated in the plot. *p*-value shown is obtained for beta cells from a multiple linear regression model considering age, sex and bod mass index (BMI) as covariates, *i.e.* model hb1ac ∼ Constant + fractions of beta cells + Age + Sex + BMI.

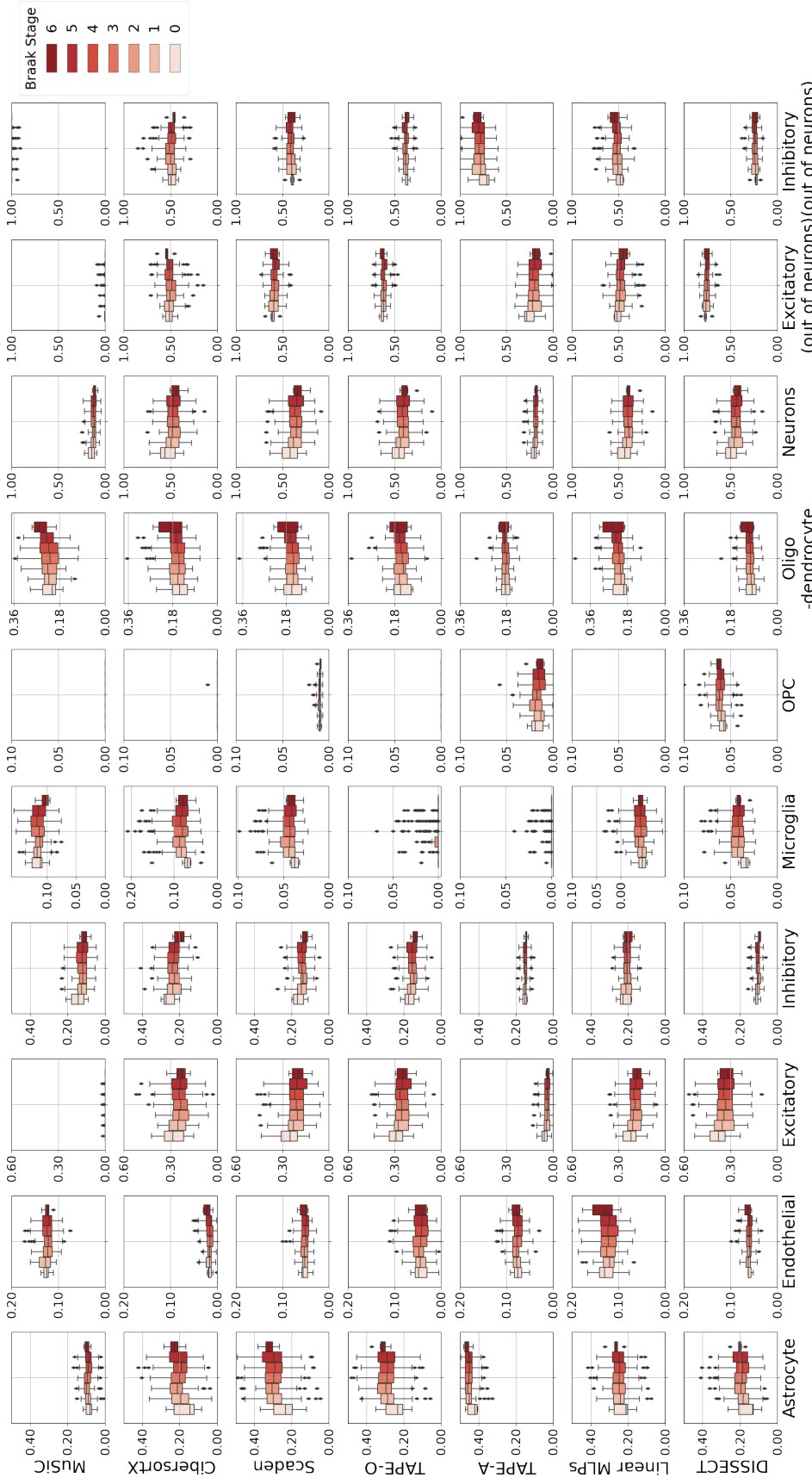

Figure 8: Fractions of cell types in 463 bulk RNA-seq samples from ROSMAP Alzheimer's Disease cohort for whom corresponding Braak stages are available. Allen Brain Atlas is used as reference single-cell data (Table 15). Rows indicate methods and each column is a cell type. *OPC*: Oligodendrocyte Precursor Cells. Excitatory and Inhibitory are two neuron subsets. Last two columns show fractions of excitatory and inhibitory neurons out of total neuronal content.

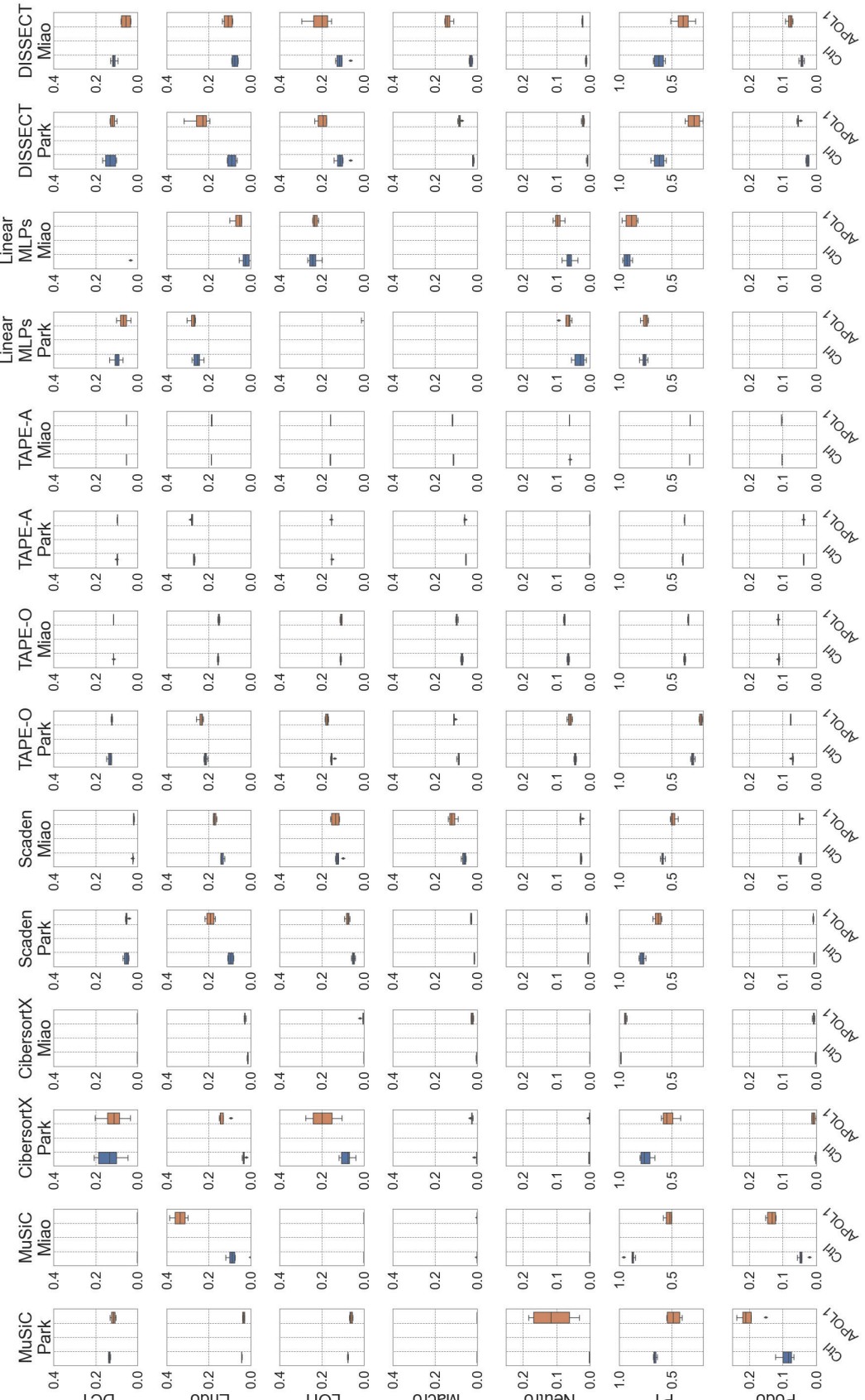

Figure 9: Estimated fractions of different cell types in 10 bulk RNA-seq samples from *GSE81492* (Ctrl: Control mice, n=6 and APOL1: Apolipoprotein L1 transgenic mice, n=4) mice using two separate single-cell reference datasets (*Park* and *Miao*). Corresponding Methods and reference datasets for each plot are indicated in the title. Each row corresponds to a cell type. DCT: Distal convoluted tubule, Endo: Endothelial cells, LOH: Loope of Henle, Macro: Macrophages, Neutro: Neutrophils, PT: Proximal Tubule, Podo: Podocytes

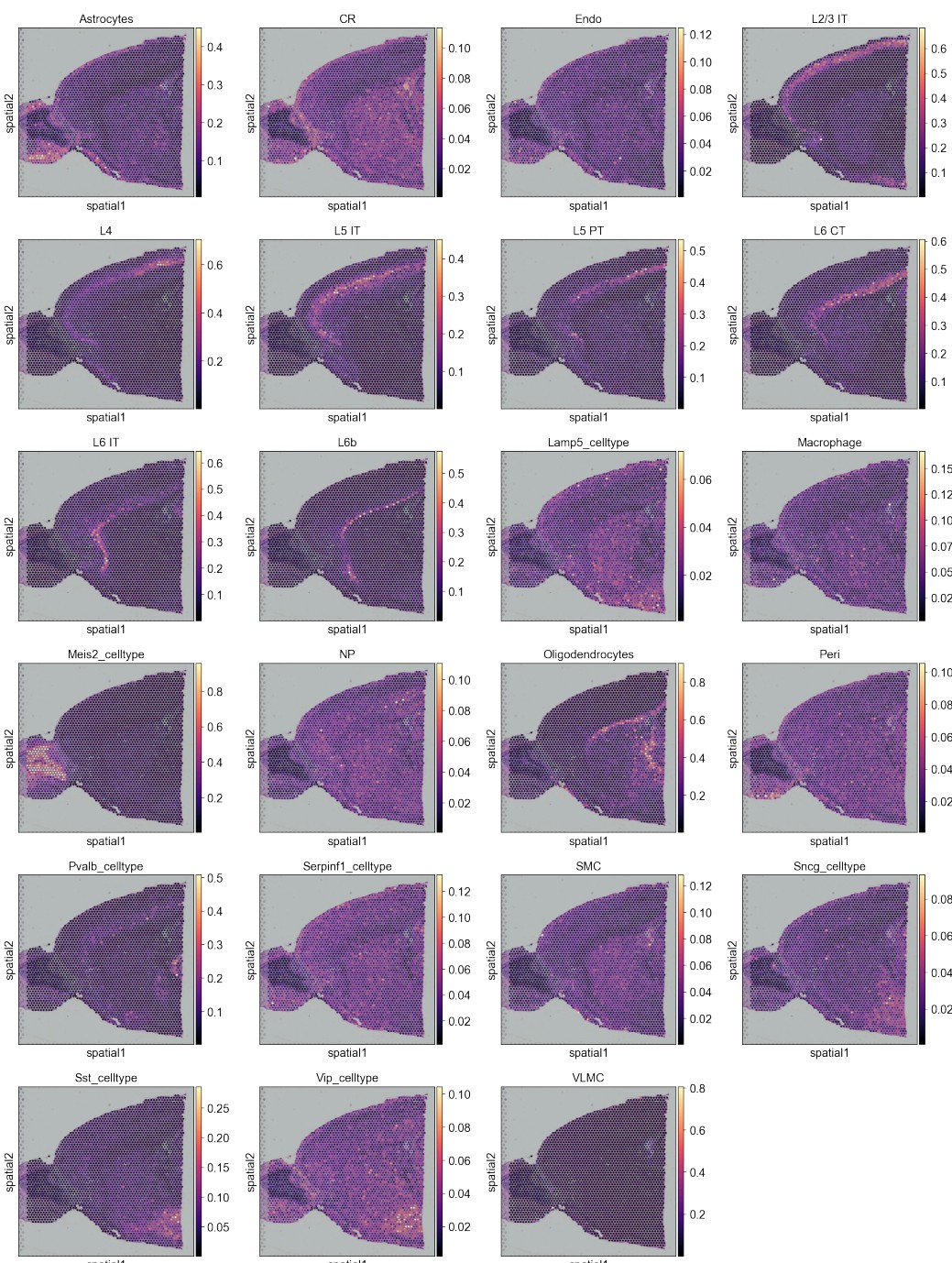

Figure 10: Fractions of all cell types shown on H&E image. Each dot corresponds to a spot sequenced in the tissue slice.

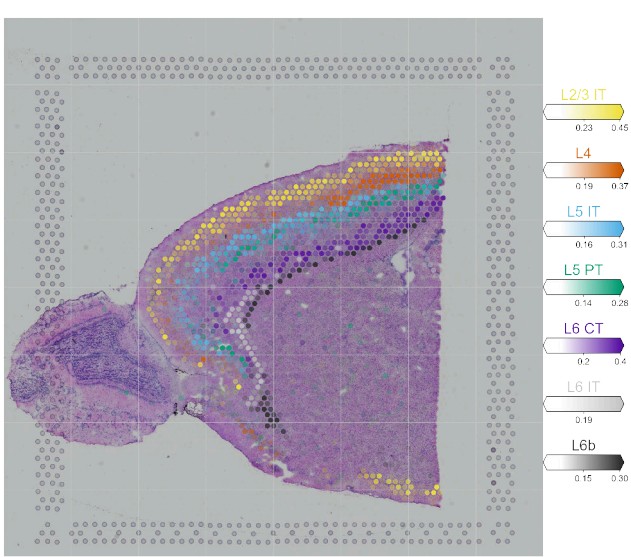

Figure 11: Cortical neurons overlaid on H&E image. The legend shows fractions.

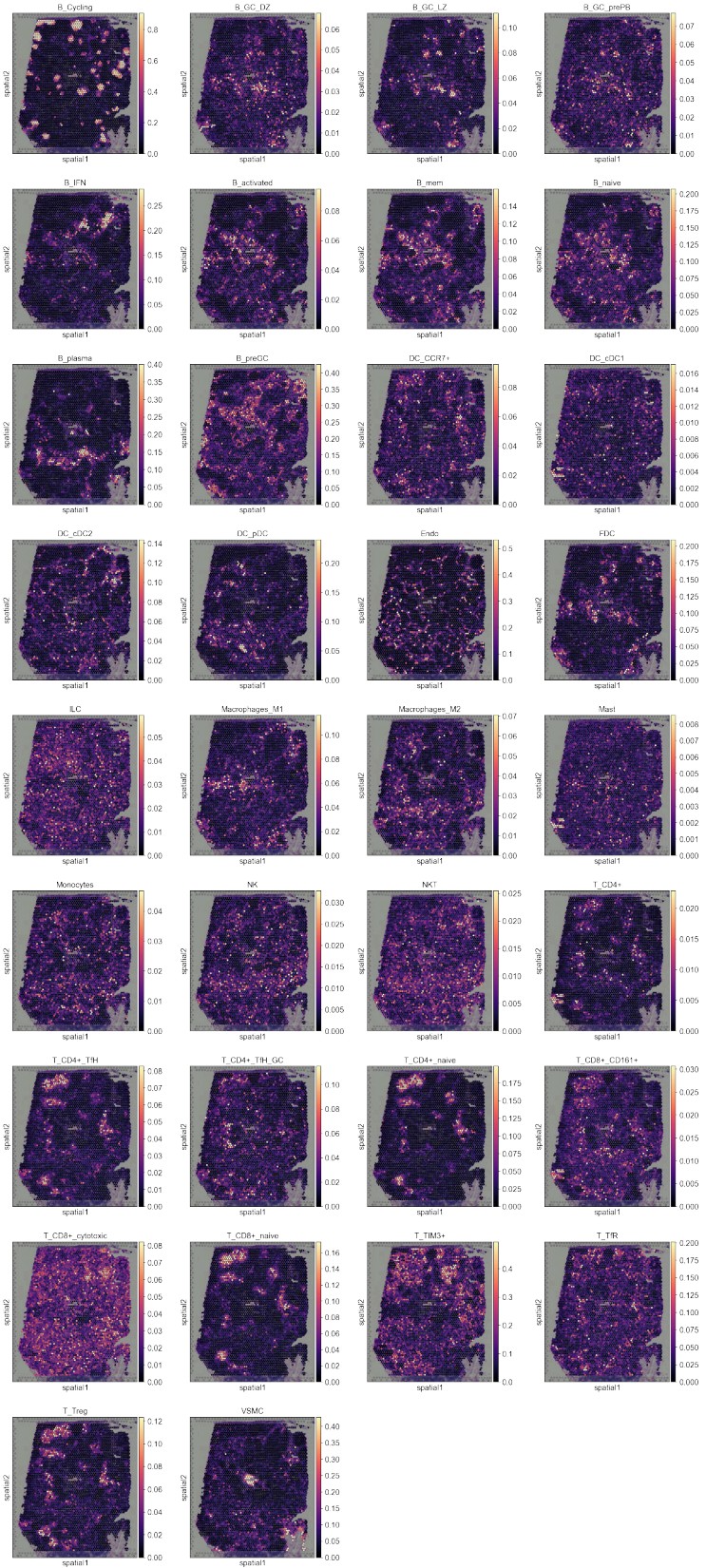

Figure 12: Estimated fractions of 34 granular level cell types overlaid on H&E image of a lymph node tissue slice.

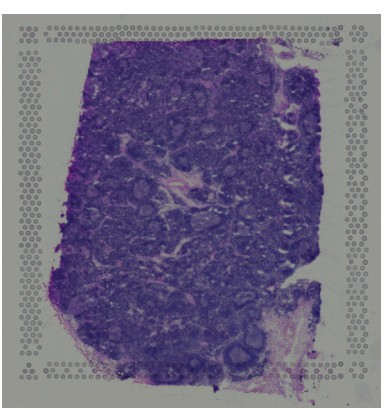

Figure 13: H&E image of lymph node tissue slice.

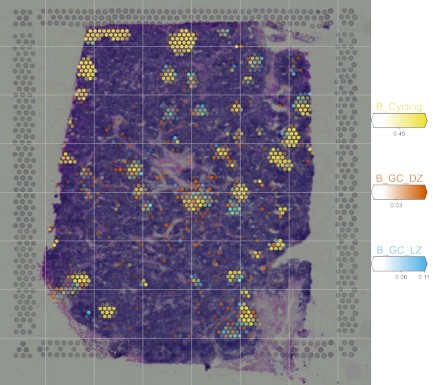

Figure 14: Fractions of Cycling and light zone (LZ) and dark zone (DZ) Germinal center B cells expected to be present in germinal centers.

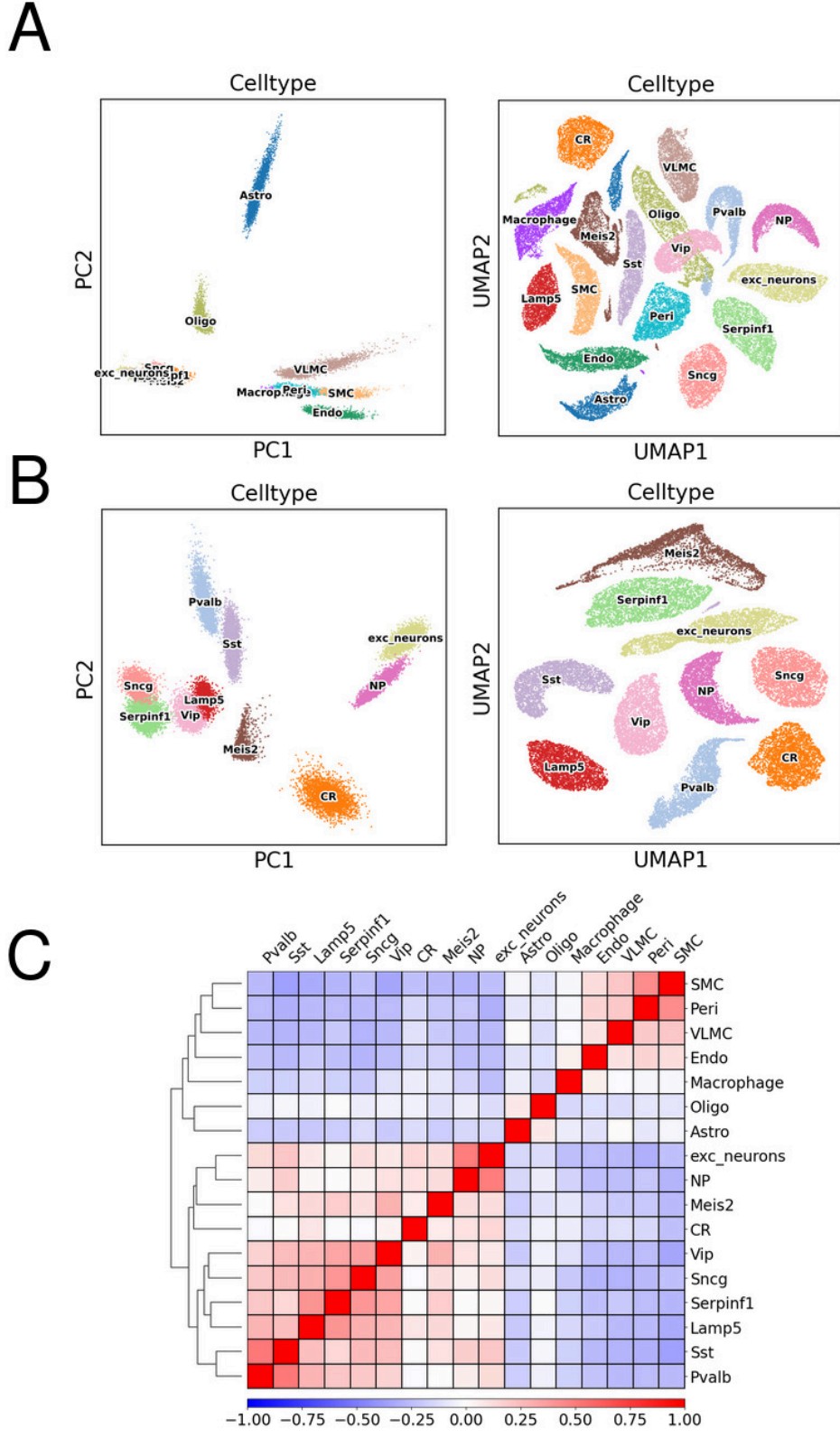

Figure 15: **A** PCA and UMAP embeddings of estimated gene expression profiles. **B**: PCA and UMAP embeddings computed on neuronal clusters. **C**: Clustered matrix showing Pearson correlation between each pair of cell types.

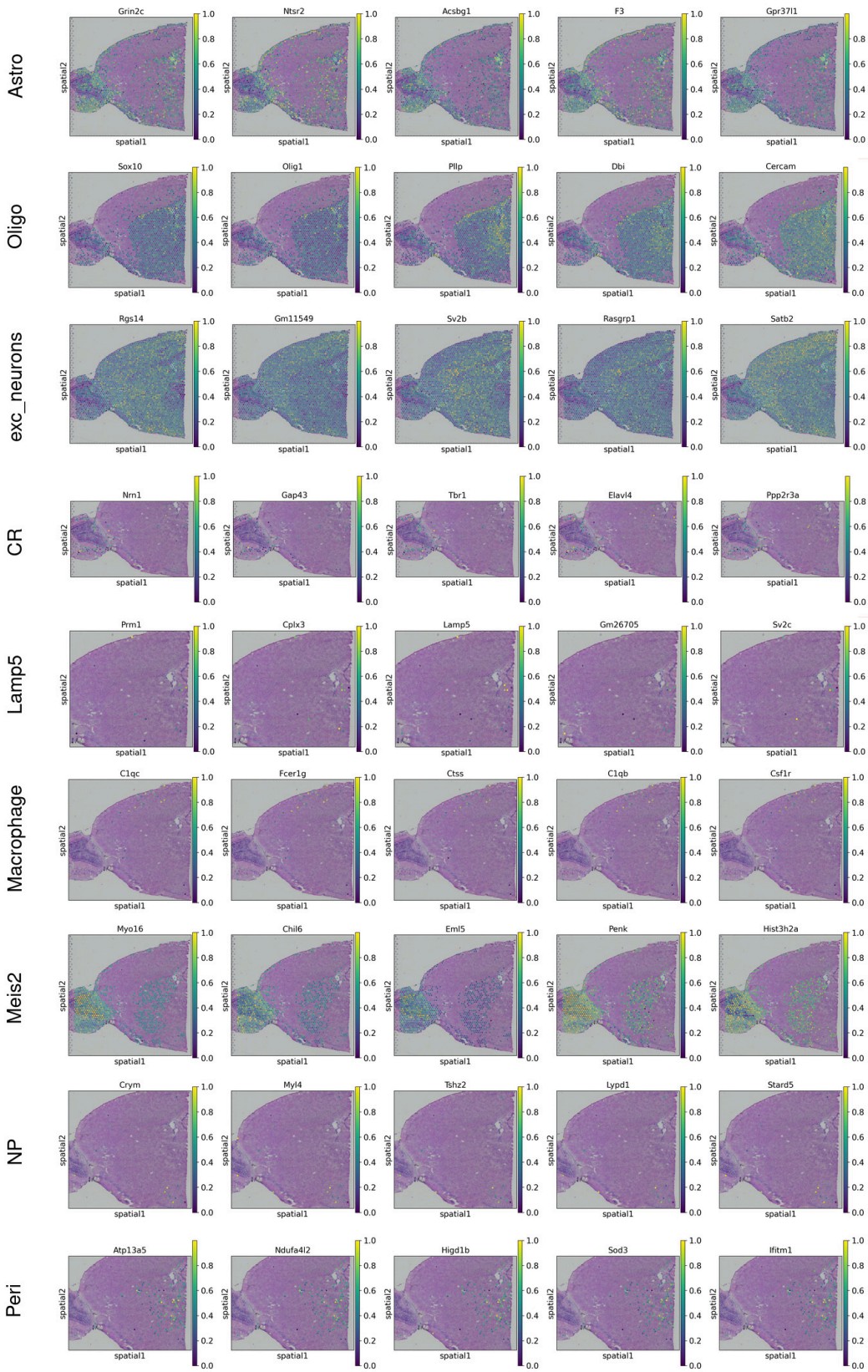

Figure 16: Scaled expression of top five DE genes for cell types shown over the H&E tissue slide. Rows indicate cell types.

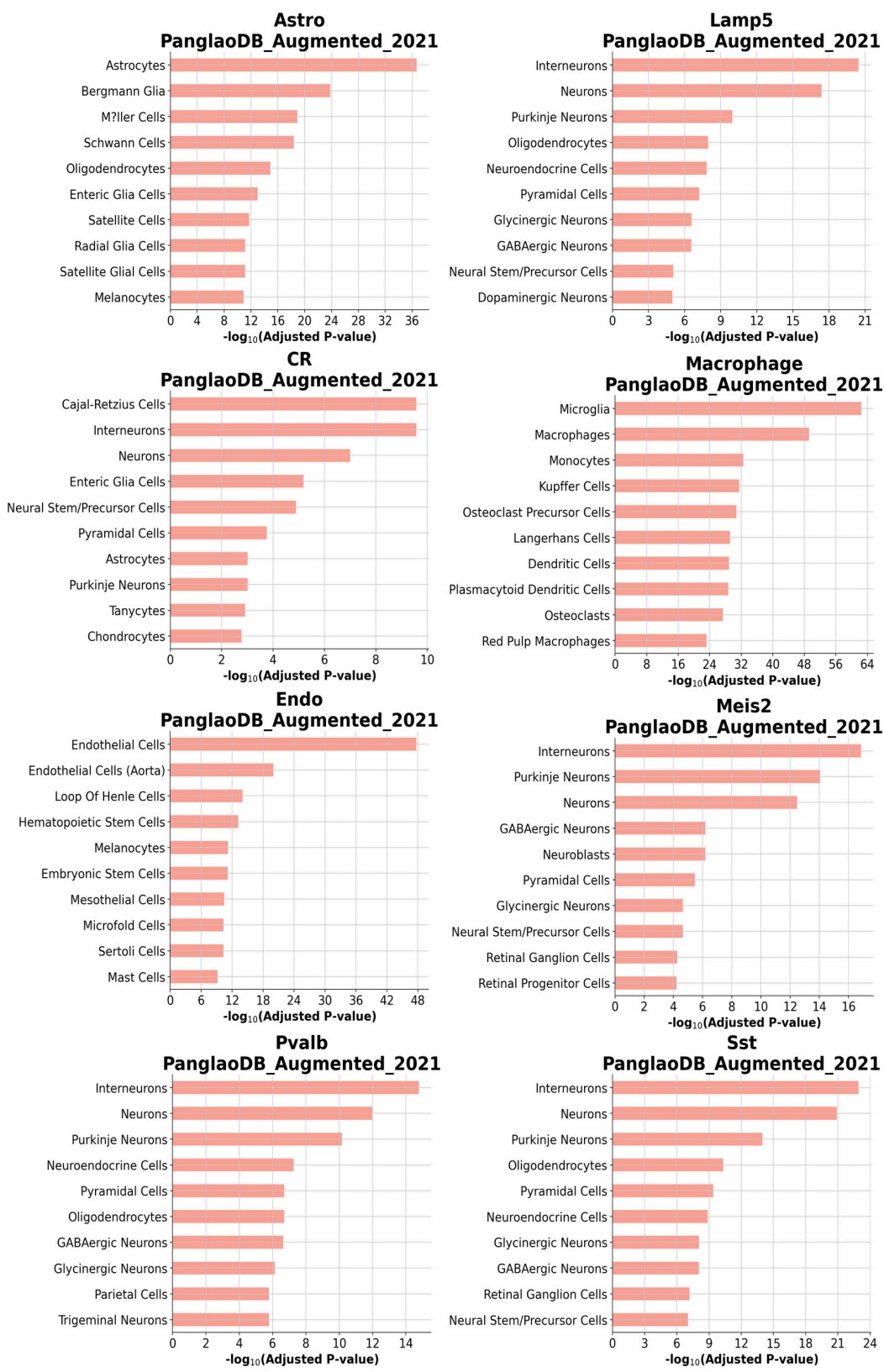

Figure 17: Plots showing gene set enrichment results of each cell type. For each cell type, the DE genes were selected with adjusted value cutoff of 0.01 and absoluted $log_2$ fold change cutoff of 1. *PanglaoDB Augmented 2021* gene sets were used as background.

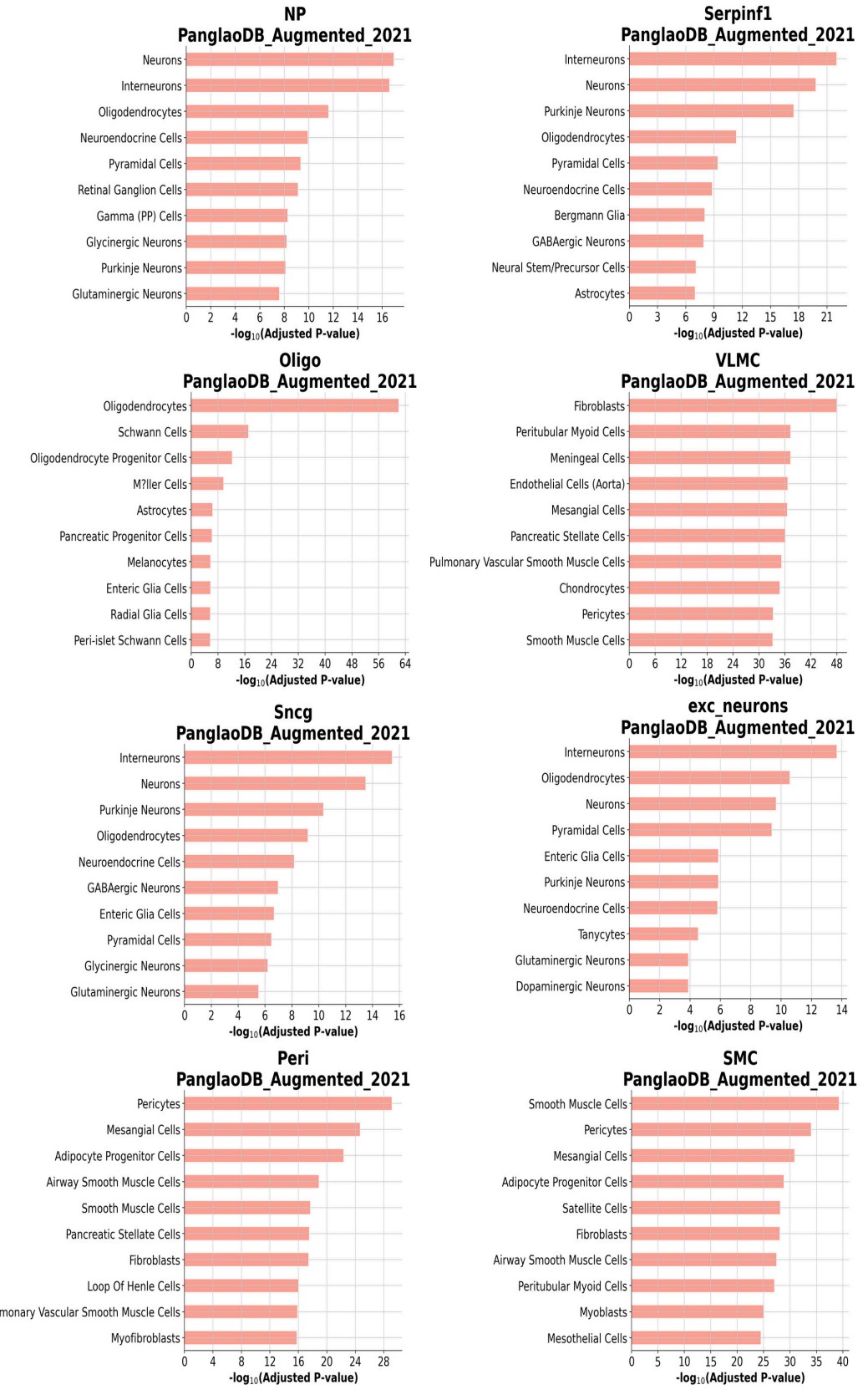

Figure 18: Continued from Figure 17.

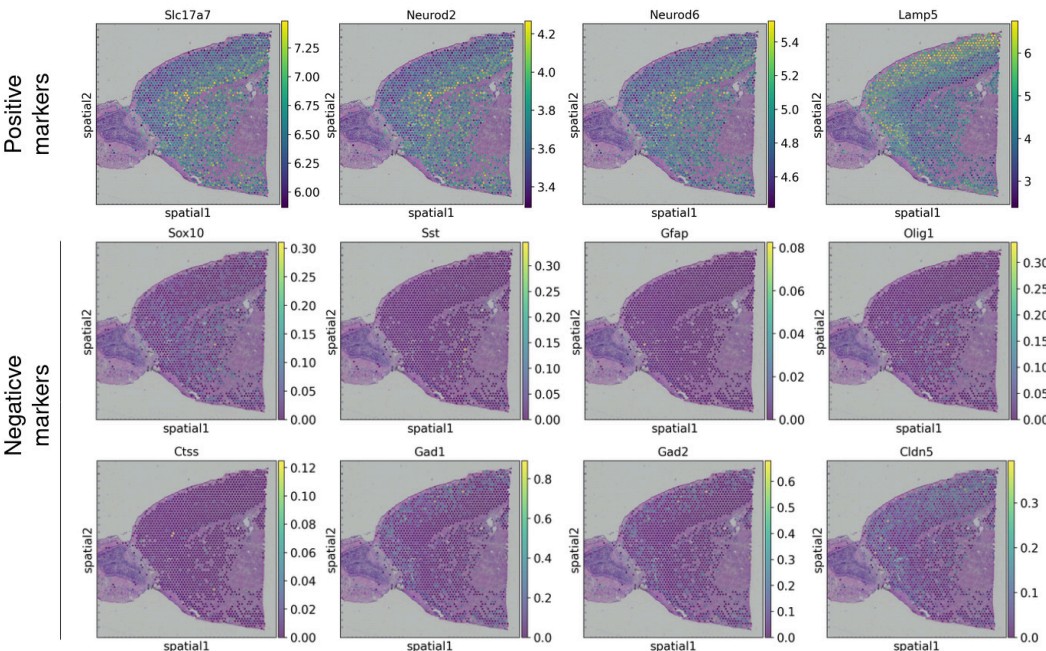

Figure 19: Evaluation of known positive and negative marker genes for excitatory neurons. The negative marker genes are the genes highly expressed on other cell types compared to excitatory neurons. *Sox10, Olig1*: Oligodendrocytes, *Sst*: Sst neurons, *Gfap*: Astrocytes, *Ctss*: Microglia, *Gad1, Gad2*: Inhibitory neurons, *Cldn5*: Endothelial cells.

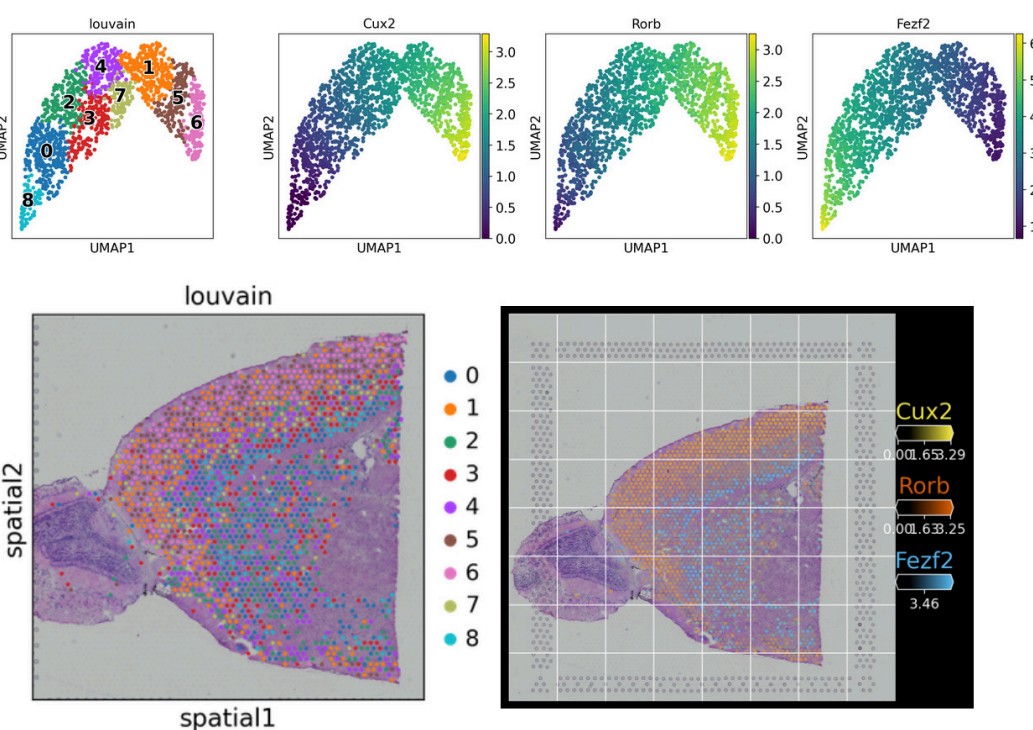

Figure 20: Top: (From left to right) Louvain clustering on estimated gene expression on excitatory neurons. $log_2$ gene expression of *Cux2*, *Rorb* and *Fezf2*. Bottom left: Louvain clusters identified on estimated excitatory neurons visualized on top of H&E slide and Bottom right: Expressions of *Cux2*, *Rorb* and *Fezf2* in excitatory neurons jointly visualized over H&E slide.

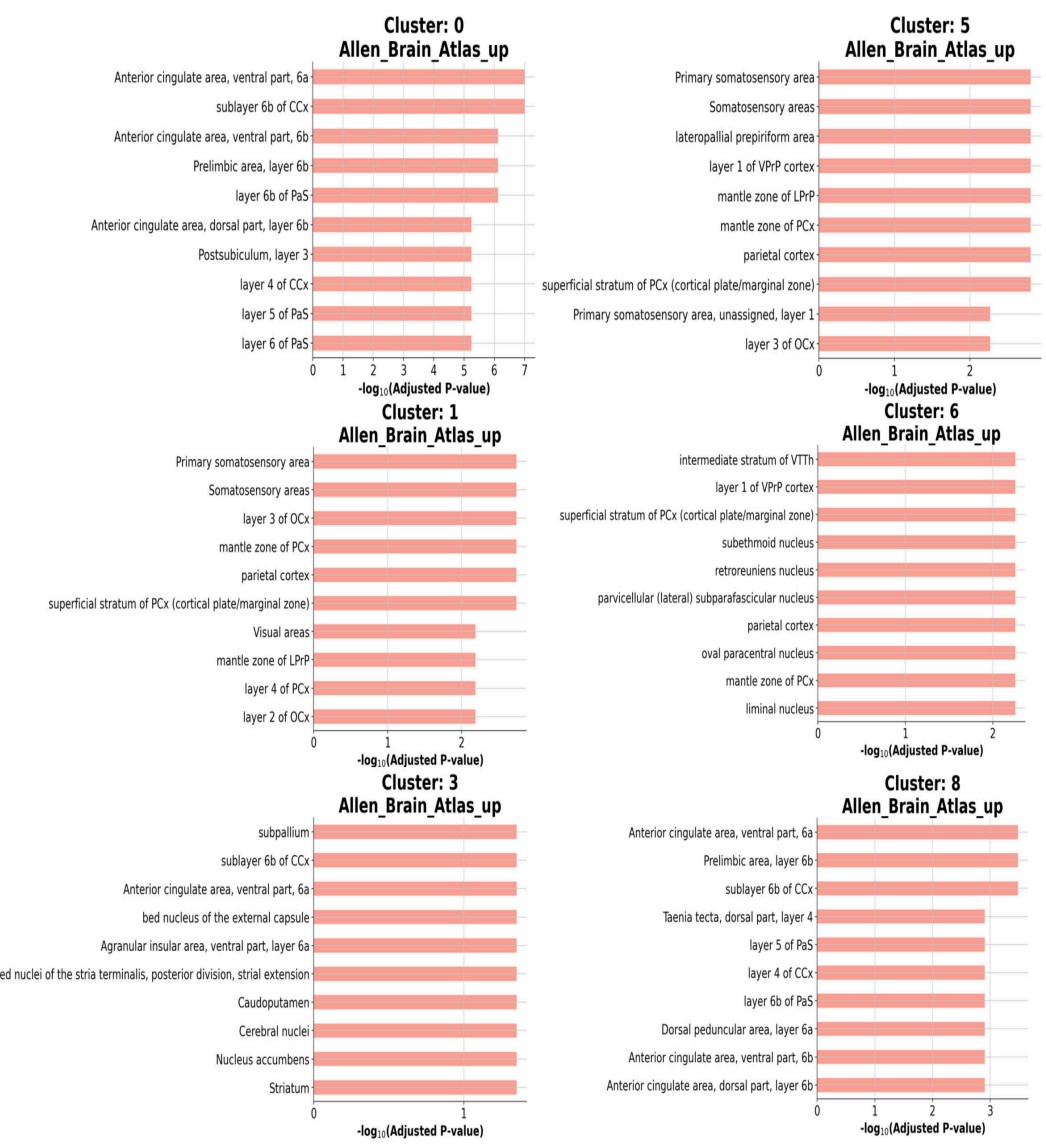

Figure 21: Gene set enrichment of DE genes computed for each louvain cluster on excitatory neurons. Enrichr was used to perform gene set enrichment of each DE gene set with *Allen Brain Atlas Up* gene sets as background. The *x*-axis indicated $-log_{10}$ adjusted *p-value*. A higher value indicates greater significance. The *y*-axis lists the enriched gene sets ordered by decreasing significance.

