# OpenReview forum: "Semi-supervised consistency regularization for accurate cell type fraction and gene expression estimation"
_ICLR.cc/2023/Conference — Submitted to ICLR 2023_

### Official Review · Reviewer_qKLA · 2022-10-22

**Confidence:** 4
**Correctness:** 4
**Technical Novelty And Significance:** 2
**Empirical Novelty And Significance:** 2
**Recommendation:** 3

**Clarity, Quality, Novelty And Reproducibility:**

- As mentioned above, there is some ambiguity about how the cell-type estimation was performed. It is stated that the PBMC8k dataset was used as the single-cell reference dataset for deconvolving the bulk RNA-seq data in Table 1. This produced the results in Table 4. Were the other scRNA-seq datasets not used then for DISSECT or any other method to produce Table 4? Was this then the only dataset used for the MuSiC? Again, if so, this would be a disadvantage for MuSiC, since it is designed for multi-sample data.
- The incorporation of the consistency regularization, while sensible and a good consideration, is not in itself a significant algorithmic advance. A more extensive evaluation would help overcome this weakness.
- I do commend the authors on the clarity of their work and believe it has potential if further developed.

**Details Of Ethics Concerns:**

No ethical concerns.

**Strength And Weaknesses:**

I believe the authors are pursuing a good research direction within the field of RNA-seq deconvolution, with the incorporation of a consistency loss into training, and this problem is important for biological studies, but this field has been studied extensively and to make a contribution to it, a more thorough investigation is needed.

Strengths:
- The problem of RNA-seq deconvolution is of great importance for many biological studies.
- Adding a consistency loss is a good insight for addressing the issue of differing characteristics between scRNA-seq and bulk RNA-seq.
- The authors are aware of other primary competing methods in the field of cell-type deconvolution.
- The authors have clearly laid out the logic of their reasoning about the consistency loss and the math details are explained clearly and are easy to follow.
- Overall, the paper is well-written.

Weaknesses:
- As mentioned, there has been extensive work toward solving the problem of deconvolution. The authors have only applied their method to datasets of PMBCs, and if I understand their training procedure correctly for cell-type estimation, only used the PBMC8k dataset as the single-cell reference (or did they use all 4 from Table 1?). Even so, these datasets are from one individual each (per the description in Table 1), thus the datasets are limited in their ability to capture possible person-to-person or technical variation. Thus, it could be that their advantage comes largely from this limitation. In comparison, MuSiC compared to several different tissue types and used scRNA-seq data of 10-20 individuals. To make a compelling case, the authors need to apply to more tissue types and use a more diverse set of reference single-cell datasets. (The authors include results for some other types of data in the Appendix, but need to provide quantitative comparisons and an explanation of the results.)
- In particular, the authors should consider doing an evaluation that exactly compares to some of those reported in MuSiC or (Avila Cobos et al., 2020); this would give greater confidence that their evaluation matches what is reported in those works.
- Furthermore, as was shown in the benchmarking study of (Avila Cobos et al., 2020), the normalization procedure can have dramatic effects on the results of deconvolution. The authors comment in the Appendix C.3 that their procedure is standard for Scaden, but given the potential dramatic impact, more reasoning should be provided for the other methods as well, especially in light of the analysis of (Avila Cobos et al., 2020).
- The incorporation of the consistency regularization, while sensible and a good consideration, is not in itself a significant algorithmic advance, at least for the scope of ICLR.
- There are other methods that are not referenced: Bisque (Jew et al., Nature Communications 2020) and DWLS (Tsoucas et al., Nature Communications 2019). The authors are correct that MuSiC and Cibersort are the main established methods, but still some reference should be made to these methods to distinguish theirs and a comparison against them would strengthen their work.

**Summary Of The Paper:**

The authors present DISSECT, a method for estimating cell fraction and gene expression from bulk RNA sequencing data. The primary innovation in DISSECT is the introduction of a regularization term during training that encourages similar expression profiles between simulated bulk data (drawn from scRNA-seq data) and real bulk RNA-seq data. A multilayer perceptron is used to estimate cell-type proportions and an autoencoder is used to estimate cell-type specific gene expression profiles. The authors evaluate their method on several real bulk RNA-seq datasets of PBMCs.

**Summary Of The Review:**

The current state of the evaluation and its limitations, as mentioned above, are the primary weaknesses of the work in its current state. A more extensive evaluation, following the analysis done in these other works in deconvolution, would greatly strengthen the paper. But in its current state, it is not ready for publication.

---

> ### Author Response · Authors · 2022-11-18
> **Thank you and our response (1 of 3)**
>
> We would like to thank the reviewer for the detailed feedback and the many helpful comments and suggestions. We have addressed all points raised and believe that our revised manuscript has gained scientific value thereafter.
>
> -----
>
> **(1) Reviewer:** “In particular, the authors should consider doing an evaluation that exactly compares to some of those reported in MuSiC or (Avila Cobos et al., 2020); this would give greater confidence that their evaluation matches what is reported in those works.”
> “The authors include results for some other types of data in the Appendix, but need to provide quantitative comparisons and an explanation of the results.”
> “To make a compelling case, the authors need to apply to more tissue types and use a more diverse set of reference single-cell datasets.”
>
> **Authors:** We completely agree with the suggestion to include more datasets for a more reliable estimate of DISSECT’s performance and we have included 2 new datasets in the revised manuscript.
>
> We would like to point out, however, that this work focuses on the deconvolution performance on real bulk RNA-seq data with ground truth fraction information. Prior works, and our own observations, indicate that performance evaluations on simulated data are overly optimistic and heavily underestimate the influence of the bulk RNA-seq bias and/or domain gap (see e.g. Scaden manuscript). The purpose of the consistency loss is to amend this bias, which is mostly absent in simulated data. Since the benchmark article of Avila Cobos focuses almost exclusively on simulated data, we only use the single real bulk RNA-seq dataset used in the Avila Cobos et al article, GSE107572, in our work.
>
> MuSiC is a great work and has contributed tremendously to utilization of single cell RNAseq to deconvolve bulk RNAseq. In fact it was one of the earliest methods to do so. However sample-level comparisons made in MuSiC relied also on simulations. The two other experiments included in MuSiC involve one Pancreas (Healthy vs Type 2 diabetes - T2D) and three Kidney bulk RNAseq datasets (chronic kidney disease mouse models). In those experiments the knowledge from literature was treated as ground truth, as indicated in Appendix Table 10.  It should be noted that literature ground truths are general in that they do not tell about a sample but they tell a global influence of a certain phenotype. The common knowledge may also change such as shown in Lawler et al (2017) where the authors observed that the drop in beta cell fractions in T2D patients may not be as significant as expected previously. Nonetheless the trends likely remain similar.
>
> The pancreas dataset (Fadista et al, 2016) and one of the kidney datasets (Backermann et al, 2019) used in MuSiC are already included in our study. Previously, we included multiple single-cell references to observe the change in deconvolution results when the reference single-cell dataset is changed (Appendix E). To do so, we subsetted these multiple single-cell references to include the same cell types. However, following your comments, we have separated these experiments into two groups: **Appendix E.2**, where we follow the identical reference datasets as used in MuSiC and evaluated on the aforementioned pancreas and kidney bulk RNAseq. **Appendix E.3**, where we compare the performance across multiple single-cell references. Briefly, in Appendix E.2, we observe that all methods result in correct trends as per literature. While in Appendix E.3, we observe that all methods produce different results when the single-cell reference dataset is swapped, with DISSECT producing least variation exemplifying its robustness. Further, we have now made an attempt to clarify Appendix E clearly and have provided explanations for each experiment.
>
> Our manuscript includes both real PBMC datasets evaluated in Scaden, an additional four PBMC bulk RNAseq datasets, a brain RNAseq, and both real tissue datasets as evaluated in MuSiC (pancreas and kidney).

---

> > ### Author Response · Authors · 2022-11-18
> > **our response (2 of 3)**
> >
> > ... **continues from the end of the previous comment**
> >
> > However, We still wanted to address your concern further. We have now added two more experiments: Mammary gland bulk RNAseq (n=2) and Covid Lung bulk RNAseq (n=17). We obtained the mammary gland dataset from Dong et al, 2021 and the COVID lung bulk RNAseq from Delorey et al, 2021. For these datasets, flow cytometry information is unavailable but both these datasets consist of paired reference scRNA-seq and bulk RNA-seq and as such allow for the quantification of cell type fractions, albeit with less confidence as compared to flow cytometry. We use the scRNAseq as reference and deconvolve the bulk RNAseq. We present these datasets and results in **Appendix E.1** and observe that DISSECT consistently outperforms other methods. However, since scRNA-seq and bulk RNA-seq come from the same studies, all methods overall performed better than what we observed in case of comparisons with flow cytometry (Table 2).
> >
> > We firmly believe that these additional results and the extensive usage of real bulk RNA-seq datasets with ground truth information in the revised document strengthened our work considerably.
> >
> > Avila Cobos, Francisco, et al. "Benchmarking of cell type deconvolution pipelines for transcriptomics data." Nature communications 11.1 (2020): 1-14.
> >
> > Lawlor, Nathan, et al. "Single-cell transcriptomes identify human islet cell signatures and reveal cell-type–specific expression changes in type 2 diabetes." Genome research 27.2 (2017): 208-222.
> >
> > Dong, Meichen, et al. "SCDC: bulk gene expression deconvolution by multiple single-cell RNA sequencing references." Briefings in bioinformatics 22.1 (2021): 416-427.
> >
> > Delorey, Toni M., et al. "COVID-19 tissue atlases reveal SARS-CoV-2 pathology and cellular targets." Nature 595.7865 (2021): 107-113.
> >
> > -----
> >
> > **(2) Reviewer:** Furthermore, as was shown in the benchmarking study of (Avila Cobos et al., 2020), the normalization procedure can have dramatic effects on the results of deconvolution. The authors comment in the Appendix C.3 that their procedure is standard for Scaden, but given the potential dramatic impact, more reasoning should be provided for the other methods as well, especially in light of the analysis of (Avila Cobos et al., 2020).
> >
> > **Authors:** The data normalization can indeed have strong effects on the deconvolution performance, which is why we have always normalized the data in accordance with the best practices of the respective method, as published in the original works. Based on your suggestion, we have now detailed the normalization procedure for each method in Appendix C.3. For DISSECT, we perform CPM normalization to make sure that mRNA expression of each gene in each sample is measured at the same scale i.e. out of a million total mRNA expression count per cell. This allows cross sample comparison. $log_2(1+CPM)$ is performed to reduce the skewness of the data (Leuken et al, 2019).
> >
> > Lastly, we would like to agree with the reviewer that it would be interesting to understand which normalization techniques could further improve deconvolution models on real data (not simulated), which could be the focus of a future manuscript. Here we would like to highlight the fact that consistency regularization is significantly improving deconvolution, most probably across different algorithmic approaches.
> >
> > Luecken, Malte D., and Fabian J. Theis. "Current best practices in single‐cell RNA‐seq analysis: a tutorial." Molecular systems biology 15.6 (2019): e8746.
> >
> > -----
> >
> > **(3) Reviewer:**  “There are other methods that are not referenced: Bisque (Jew et al., Nature Communications 2020) and DWLS (Tsoucas et al., Nature Communications 2019). The authors are correct that MuSiC and Cibersort are the main established methods, but still some reference should be made to these methods to distinguish theirs and a comparison against them would strengthen their work.”
> >
> > AR: This is a valid comment and we have now referenced Bisque and DWLS in the State of the Art Section of the revised manuscript. Unfortunately, it is virtually impossible to reference all of the great deconvolution methods that have been published in the last five years, as they are just too many. As is the case for most publications in the field, we have thus tried to focus on the ones which worked best in our hands (or in review articles), which were CSx, Scaden, and MuSIC. We hace referenced DWLS and Bisque by modifying Section 3 to read “... Here, we focus on MuSiC, CSx, Scaden and TAPE although here are several additional methods such as DWLS (Tsoucas et al., 2019) and Bisque (Jew et al., 2020) etc., because both MuSiC and CSx are …“

---

> > > ### Author Response · Authors · 2022-11-18
> > > **our response (3 of 3)**
> > >
> > > **(4) Reviewer** “As mentioned, there has been extensive work toward solving the problem of deconvolution. The authors have only applied their method to datasets of PMBCs, and if I understand their training procedure correctly for cell-type estimation, only used the PBMC8k dataset as the single-cell reference (or did they use all 4 from Table 1?). Even so, these datasets are from one individual each (per the description in Table 1), thus the datasets are limited in their ability to capture possible person-to-person or technical variation. Thus, it could be that their advantage comes largely from this limitation. In comparison, MuSiC compared to several different tissue types and used scRNA-seq data of 10-20 individuals. To make a compelling case, the authors need to apply to more tissue types and use a more diverse set of reference single-cell datasets.”
> > >
> > > **Authors:** This is a great comment which we addressed in the revised version of the manuscript. We started our work based on a previous deconvolution work Scaden and as such started with the datasets used there. In Scaden, only PBMC8k  was used as a reference for MuSiC. However, following your very valid concern, we have now added results of MuSiC in multi sample settings in the results Table 4, by taking all single cell PBMC samples in Table 2. MuSiC improves drastically in performance but still lags far behind CibersortX, Scaden and DISSECT. We also considered adding a larger, high quality dataset with more donors and cells for MuSiC. To this end, we obtained blood data from Immune Cell Atlas (Conde et al, 2022) which consists of six donors (Appendix Table 7). We evaluated MuSiC using this atlas-level data and saw another increase in performance, however, still at a lesser level than CSx, Scaden and DISSECT. Both multi-sample results for MuSiC are added in Table 4.
> > >
> > > Domínguez Conde, C., et al. "Cross-tissue immune cell analysis reveals tissue-specific features in humans." Science 376.6594 (2022): eabl5197.
> > >
> > > -----
> > >
> > > **(5) Reviewer:** “The incorporation of the consistency regularization, while sensible and a good consideration, is not in itself a significant algorithmic advance, at least for the scope of ICLR.”
> > >
> > > **Authors:** We would like to humbly disagree with this assessment, at least in part. Our work deals with an important application of machine learning. As such, its objectives are different to that of a methodology focused machine learning work. This is also the reason for our submission to the area “Machine Learning for Sciences (e.g. biology, physics, health sciences, social sciences, climate/sustainability”. Further, to our knowledge machine learning application papers also fall within its scope as indicated in ICLR’s call of papers: “We consider a broad range of subject areas including feature learning, … as well as **applications in** vision, audio, speech , language, music, robotics, games, **healthcare, biology**, sustainability, economics, ethical considerations in ML, and others.”. To the wider ICLR community, it presents an existing challenging and significant problem and proposes using machine learning advancements in semi-supervised learning to approach it.
> > >
> > > Our method starts with an assumption regarding the possibility of improvement in performance with the inclusion of this regularization method. We verify this assumption empirically throughout the paper. We think that proposing online generation of mixtures of simulated and real bulk gene expression data, and regularizing the training with a consistency loss is a novel contribution in the task of cell deconvolution. It is an attempt to incorporate information from bulk RNA-seq data in a task dependent manner which has not been explored before. Further, deconvolution is not a task limited to bulk RNA-seq. It arises in several other tasks in biology and health care, namely finding origin of cells in DNA methylation, recovering cell type distribution in spatial transcriptomics, signature enrichment for omics e.g. proteomics, microRNA, and metabolomics, for instance. Beyond omics, deconvolution falls in the broader task of semi-blind source separation. Our work demonstrates how utilizing domain knowledge in this context could improve model performances, in general. Depending on the field, the consistency loss may take different forms, or another form of source-mixture simulations would be needed.
> > >
> > > To our knowledge use of gene expression from bulk RNAseq data in either construction of the signature matrices (for non deep learning based methods) or during the learning process (for deep learning based methods) has not been explored before and we have gone through important works between 2009 and 2022: https://www.notion.so/anonymousworkspace/Literature-summary-43fb613dcfd1435da0714b22e55ce09f. Our work is an attempt to address this. As you pointed out, it is a sensible consideration.
> > >
> > > -----
> > > We hope that we were able to satisfy your concerns. We welcome and highly value your feedback.

---

> ### Author Response · Authors · 2022-12-06
> **Follow-up**
>
> We thank the reviewer once again for the feedback. We are looking forward to the assessment of our responses. We would be happy to continue the discussion.

---

### Official Review · Reviewer_fXFv · 2022-10-23

**Confidence:** 5
**Correctness:** 4
**Technical Novelty And Significance:** 3
**Empirical Novelty And Significance:** 3
**Recommendation:** 3

**Clarity, Quality, Novelty And Reproducibility:**

The quality, clarity and originality of the work are OK.


**Strength And Weaknesses:**

Strength

- The proposed model was extensively tested against existing models using various datasets.
- The proposed model showed superior preformance among compared.

Weaknesses

- Whole contents of this paper is focusing on a particular task in cell biology. Very domain specific.
- It is unclear how the model was trained. Which dataset was used?


**Summary Of The Paper:**

The authors propose a new model for cell deconvolution task by introducing consistency regularization.


**Summary Of The Review:**

The authors propose a new model for cell deconvolution task by introducing consistency regularization. The paper really focusing on a particular task in cell biology thus more specific venue is suitable.

---

> ### Author Response · Authors · 2022-11-18
> **Thank you and our response**
>
> First of all we would like to thank the reviewer for the valuable comments and several positive remarks about DISSECT’s performance, the extensive testing performed, and the technical novelty and significance of the work. We understand that the main point of critique is the narrow domain this work relates to, which we would like to rebut below.
>
> -----
>
> **(1) Reviewer:** “Whole contents of this paper is focusing on a particular task in cell biology. Very domain specific.”
>
> **Authors:** Indeed, we have focused the evaluation on a particular task in cell biology but we believe that the impact goes well beyond this particular task and cell biology.
>
> Importantly, deconvolution in cell biology is not applicable only to a single task. It arises in several other tasks in cell biology, namely finding origin of cells in DNA methylation, recovering cell type distribution in spatial transcriptomics, signature enrichment for omics e.g. proteomics, microRNA, and metabolomics, for instance. In this context we would like to note that cell deconvolution is already in use in medical settings, making this work relevant to the fields of biology and health care.
>
> Furthermore, deconvolution falls within the broader task of semi-blind source separation. Our work demonstrates how utilizing domain knowledge in this context can significantly improve model performances beyond a focused application in cell biology. Of Course depending on the field, the consistency loss may take different forms, or another form of source-mixture simulations would be needed.
> Lastly, the scope and reviewer guidelines of ICLR state that “We consider a broad range of subject areas including feature learning, … as well as applications in vision, audio, speech, language, music, robotics, games, healthcare, biology, sustainability, economics, ethical considerations in ML, and others.”. To the wider ICLR community, it presents an existing challenging and significant problem and proposes using machine learning advancements in semi-supervised learning to approach it.
>
> We hope that we could convince the reviewer, at least in part, that our work is of sufficiently broad interest to be suitable for ICLR.
>
> -----
>
> **(2) Reviewer:** Which dataset was used?”
>
> **Authors:** Following your well founded question, we now added a comprehensive Section about data usage in the revised manuscript Section 4. The datasets listed deal with the experiments that concern the main part of the paper. As now noted in Section 4.3, we provide information on other datasets in appendix E. “In this Section, we present results on experiments detailed in Section 4.1. Additional experiments and results on datasets without corresponding flow cytometry fractions are presented in Appendix E. There we utilize validated biological hypotheses to evaluate DISSECT against competing methods.”
>
> -----
>
> **(3) Reviewer:** “It is unclear how the model was trained.”
>
> **Authors:** We have improved the model training details in the revised manuscript in accordance with your question. The architecture, hyperparameters and training details concerning the models are specified in Sections 2.2.1 and 2.2.2. The detailed quality control, simulations, and pre-processing steps are provided in appendices C.1, C.2 and C.3, respectively, and are referenced in the main text. The revised manuscript now also contains an expanded appendix C.2 to include preprocessing information for all methods.
>
> -----
>
> We hope that we have answered your questions and have been able to satisfy some of your concerns. We highly value your feedback.

---

> ### Author Response · Authors · 2022-12-06
> **Follow-up**
>
> We thank the reviewer once again for the feedback. We are looking forward to the assessment of our responses. We would be happy to continue the discussion.

---

### Official Review · Reviewer_w6RR · 2022-10-24

**Confidence:** 4
**Clarity, Quality, Novelty And Reproducibility:** Authors need to clarify some details …
**Correctness:** 3
**Technical Novelty And Significance:** 3
**Empirical Novelty And Significance:** 2
**Recommendation:** 5

**Strength And Weaknesses:**

The manuscript was well written and included major players in the field of celltype deconvolution to compare in the computational experiments. The shared code repository was helpful to better understand the algorithm. There are some experiment details that need to be clarified as follows.

Major Concerns

1.	(section 4.3.2) The experiments estimating celltype-specific gene expression was only conducted on the simulated bulk RNA-seq data. This is insufficient to evaluate how DISSECT behaves in practice. One possible source of verification data can be imaging-based spatial transcriptomics data [1], as authors have tried applying DISSECT on the spatial transcriptomic data.

2.	(section 3) Authors constructed the Linear MLPs removing the consistency loss which may be not a fair comparison. I am wondering how Linear MLPs would behave if the consistency loss was incorporated.

3.	(section 2.2.1) The first training phase was purely based on the simulated examples. I am curious to see what the performance of DISSECT is right after this simulation-based training phase. This can be seen as an ablation test to check how mixing the training data with training data helps improve the deconvolution performance.

4.	I didn’t find descriptions on how the training/validation/test sets were split for each dataset. Could you elaborate on this?

Minor Concerns

1.	Scaden [2] included both SDY67 (called PBMC1) and Monaco (named PBMC2) for comparison in their manuscript. Their reported correlations were different compared to those reported in Table 3. I am wondering what could possibly lead to this difference.


[1] Cable, Dylan M., et al. "Cell type-specific inference of differential expression in spatial transcriptomics." Nature methods 19.9 (2022): 1076-1087.

[2] Menden, Kevin, et al. "Deep learning–based cell composition analysis from tissue expression profiles." Science advances 6.30 (2020): eaba2619.


**Summary Of The Paper:**

Authors proposed DISSECT as a novel deconvolution framework estimating both celltype fractions and celltype specific gene expression. DISSECT leveraged both the training data with ground-truth proportions and simulated data based on scRNA-seq reference. It can be seen as an augmentation strategy which mixes the training data with synthetic data or a regularization framework that solves the domain shift problem between the training domain and reference (scRNA-seq) domain. Authors completed extensive evaluations over a large collection of datasets and compared DISSECT with other competitive methods. Authors highlighted the superior performance of DISSECT and provided a possible extension on applying DISSECT on the spatial transcriptomic data which was a new application scenario. 

**Summary Of The Review:**

The manuscript is not ready yet to be presented in its current form. Concerns raised above need to be addressed.


Response to authors

Thank you for the detailed response. Most of my concerns have been addressed properly. I still have two minor comments:
1. I am confused that the consistency loss will be zero for linear MLP as I follow the definition of consistency loss in equation 6.

2. Thanks for including appendix table 13 and 14. It will be great if they can be merged to table 3 and table 4 as one separate column accordingly. Also the column name of the first field in table 13 should be updated from "Dataset" to "Metrics" or something similar.

I will keep my scores for now.

---

> ### Author Response · Authors · 2022-11-18
> **Thank you and our response (1 of 2)**
>
> We would like to thank the reviewer for the detailed feedback and the many helpful questions and comments. We have addressed all points raised and believe that our revised manuscript has gained scientific value thereafter.
>
> -----
>
> **Reviewer:** (Section 4.3.2) The experiments estimating celltype-specific gene expression was only conducted on the simulated bulk RNA-seq data. This is insufficient to evaluate how DISSECT behaves in practice. One possible source of verification data can be imaging-based spatial transcriptomics data [1], as authors have tried applying DISSECT on the spatial transcriptomic data.
>
> **Authors:** Indeed using simulations may not reflect the real world performance. In fact, it is the noted in some prior deconvolution works such as Scaden (Menden et al, 2020) and assumed in our work DISSECT that simulated data does not faithfully reflect real data, which is why we tried to benchmark on real data with ground truth information wherever possible. Evaluation of gene expression estimation only on simulations is tied to unavailability of bulk RNAseq from paired tissue and purified cell populations.
>
> However, using spatial transcriptomics (ST) is an excellent suggestion. Indeed, due to the spatial information, we can qualitatively verify whether correct genes are expressed in locations where it’s expected. Following your suggestion, we designed an experiment using the datasets we previously utilized in demonstrating the application of DISSECT in spatial transcriptomics (ST) in Appendix F. The detailed description and results of the experiment setting are provided in Appendix H in our revised manuscript.  Briefly, In this experiment, we were interested in answering two questions:
>
> 1. Does DISSECT estimates of gene expression reflect what is observed for that cell type in the literature.
>
> 2. Can DISSECT identify heterogeneity of the same cell type across samples (in this case spots) without having to pre-annotate cell type subsets. To accomplish this, we merged excitatory neuronal subsets together and labeled them as “exc_neurons”. This allows us to test whether we observe heterogeneity in excitatory neurons after estimation or not. This resulted in 17 final cell types. We also filtered our cells where the proportion of corresponding cell type is less than 1/(no. of cell types). This is reasonable as 10x Visium spots contain between 1-10 cells.
>
> The first question relates to accuracy of the predicted signature, and the second question is about whether the biological reality of the sample at hand is preserved.
>
> To evaluate our results, we looked at clustering of cell types (through PCA and UMAP embeddings), gene set enrichment, visualized marker genes and have explored capability of DISSECT to capture spatial heterogeneity of excitatory neurons.
>
> While the above experiment supports our method, we still consider the question of verification of gene expression estimation as a limitation of our study. However, this is a limitation largely dependent on the availability of paired bulk RNA-seq and purified cell type specific bulk RNA-seq, and in future we may be able to resolve it.
>
> We have added the following statement in the Section 6 (Limitations): “As stated in Section 4.3.2 we had to rely on the simulation based experiment to evaluate gene expression estimations. Nevertheless, we still explored how well DISSECT can estimate gene expression using an ST dataset (Appendix H). Further evaluations with quality ground truths will be beneficial..“
>
> Franzén, Oscar, Li-Ming Gan, and Johan LM Björkegren. "PanglaoDB: a web server for exploration of mouse and human single-cell RNA sequencing data." Database 2019 (2019).

---

> > ### Author Response · Authors · 2022-11-18
> > **our response (2 of 2)**
> >
> > **(2) Reviewer:** (Section 3) Authors constructed the Linear MLPs removing the consistency loss which may be not a fair comparison. I am wondering how Linear MLPs would behave if the consistency loss was incorporated.
> >
> > **Authors:** Indeed that would be a nice comparison, however, due to the linear nature of the consistency loss, for a linear MLP, the consistency loss will always be 0 as $f(x_1+x_2) = f(x_1) + f(x_2)$ for a linear function $f$.
> >
> > -----
> >
> > **(3) Reviewer:** (Section 2.2.1) The first training phase was purely based on the simulated examples. I am curious to see what the performance of DISSECT is right after this simulation-based training phase. This can be seen as an ablation test to check how mixing the training data with training data helps improve the deconvolution performance.
> >
> > **Authors:** Thank you for this suggestion. We have now added results of the simulation-based training phase alongside our previous ablation results in Appendix G. In brief, result post simulation based training is suboptimal compared to the final results. To reference the Appendix, we added the following text in the Section 4.3.1: “We also evaluated the output of DISSECT at the end of simulation-phase only. These results are provided in Appendix Tables 13 (Correlation) and 14 (RMSE) where simulation-based phase lags behind the full consistency-regularized training.”
> >
> > -----
> >
> > **(4) Reviewer:** I didn’t find descriptions on how the training/validation/test sets were split for each dataset. Could you elaborate on this?
> >
> > **Authors:** Indeed we did not provide any information on training, validation, and test splits in the document. The reason for this is that DISSECT’s training data is always simulated and the test dataset is always bulk RNA-seq, which obviates the need for training, validation and test splits. After selecting the best hyperparameters of the network, we used the same network for deconvolving each bulk RNA-seq. The ‘trick’ is that we do not pass the ground truth information of the bulk RNA-seq data to the network during training, we only optimize the consistency. This is why one can immediately use the trained network for inference on the same bulk RNA-seq data after training on simulated and bulk RNA-seq mixtures. For the estimation of cell type-specific gene expression on simulated datasets, we provide the information about training and test sets in Section 4.3.2 under Table 4.
> >
> > -----
> >
> > **(5) Reviewer:** Scaden [2] included both SDY67 (called PBMC1) and Monaco (named PBMC2) for comparison in their manuscript. Their reported correlations were different compared to those reported in Table 3. I am wondering what could possibly lead to this difference.
> >
> > **Authors:** You are completely correct that correlations are different for those two datasets between what is reported in Scaden (Menden et. al., 2020). The difference is due to different training datasets. As mentioned in Scaden's manuscript under Section **Robust deconvolution of bulk expression data**, “Deconvolution for all methods was performed as described in the previous Section, with the difference that data from all four PBMC scRNA-seq datasets were now deployed for Scaden training.” In other words, Scaden in the original publication was trained on four simulations from four scRNA-seq datasets, while for the other methods, different reference scRNA-seq datasets were used: PBMC8k dataset was used for MuSiC, as mentioned in Scaden's manuscript under subsection **MuSiC** (Section **Methods**), "To generate MuSiC deconvolution predictions on PBMC datasets, we used the data8k scRNA-seq dataset as reference data for MuSiC and follow the tutorial provided by the authors to perform the deconvolution." and PBMC6k dataset was used for CibersortX, as mentioned in Scaden's manuscript under subsection **CibersortX** (Section **Methods**), "All PBMC datasets were deconvolved using a GEP matrix generated from the data6k dataset ..." . In this manuscript, however, we kept the training dataset the same for all methods to maintain comparisons.
> >
> > Menden, Kevin, et al. "Deep learning–based cell composition analysis from tissue expression profiles." Science advances 6.30 (2020): eaba2619.
> >
> > -----
> > We hope that we have answered your questions and satisfied some of your concerns. We would like to thank the reviewer again for the very constructive and detailed review.

---

> ### Author Response · Authors · 2022-12-06
> **Follow-up**
>
> We thank the reviewer once again for the feedback. We are looking forward to the assessment of our responses. We would be happy to continue the discussion.

---

### Official Review · Reviewer_W86N · 2022-10-24

**Confidence:** 4
**Correctness:** 3
**Technical Novelty And Significance:** 2
**Empirical Novelty And Significance:** 2
**Recommendation:** 3

**Clarity, Quality, Novelty And Reproducibility:**

Clarity:

The paper is well organized. There are some typos (and after question, “+” in Figure 1) in the text that need to be fixed.

Novelty:

The paper does not have any methodological originality and lacks sufficient contribution.

Reproducibility:

The results seem to be reproducible.


**Details Of Ethics Concerns:**

no ethics concerns.

**Strength And Weaknesses:**

Strength:

- The authors suggested a simple regularization loss that factorizes bulk RAN-seq and estimate cell type-specific expression with limited number of cells.

- The authors reported their model performance for several RNA-seq data and demonstrated the outperformance of their regulation-based deconvolution in compared with other deconvolution methods,


Weaknesses / Concerns / Comments:

- The paper does not have sufficient contributions. There is no justification for the consistency term and it is not clear to me that why such a simple linear formulation can improve cell type fraction approximation.

- Is there sample-to-sample correspondence between $B_i$ and $B_i^{sim}$?

- While bulk measurements and simulated dataset (based on single-cell measurements) consist of different sources of noise and biological variabilities, can you justify why a linear mixture of these two cell measurements can enhance cell deconvolution?

- Is there any equation missing after the third equation, Section 2 (page 2)?

- In Section 2, 4th equation, why does the signature matrix which is a $c$ x $n$ matrix is a function of three indices, including sample id? Does not $S$ represent the expression profiles across cell types?

- Section 2.1.2, Eq. 4 should be written as $x_i s_i$, not $s_i x_i$.

- Is the number of cell types given to the model apriori? How does the deconvolution performance change in the absence of having the true number of cell types in the bulk RNA-seq data?

- In Figure 1.a, I think the first “+” operation should be “=”.

- Why do you use a variational approach to learn signature matrix? Why not just an MLP, similar to $X$ matrix estimation? Is there any plan to use the latent factors?

- How many genes are profiled in datasets in Tables 1 & 2? Are you using the same subset of genes in Eq. 3?

- I think reporting MSE (which is reported in the supplement) is more informative rather than reporting correlation between estimated variables and true ones. My suggestion is to report MSE results in the main text.


**Summary Of The Paper:**

The paper proposed DISSECT, a regularization-based deep learning approach for bulk RNA-seq deconvolution that decomposes the bulk RNA-seq matrix into a cell type fraction matrix $X$ and an expression profile $S$. The authors introduced consistency regularization which utilizes a mixture of the true bulk RNA-seq and simulated RNA-seq data. The proposed learning paradigm includes two deep networks, MLP and VAE, estimating X and S, respectively. The authors reported the model performance for multiple bulk RNA-seq datasets and showed DISSECT outperforms the competing approaches in both cell fraction and gene expression estimation tasks.


**Update after rebuttal:** I have read all reviewers' comments and the authors' responses. I still think the paper lacks enough contribution as a machine learning paper and my score remains unchanged.

**Summary Of The Review:**

I think the paper has limited contributions to the machine learning field and the manuscript is not sufficient for a ML focused venue like ICLR.

---

> ### Author Response · Authors · 2022-11-18
> **Thank you and our response (1 of 4)**
>
> We would first like to thank the reviewer for the many detailed and great comments and suggestions. We truly appreciate it.
>
> -----
>
> **(1)** **Reviewer**: “I think the paper has limited contributions to the machine learning field and the manuscript is not sufficient for a ML focused venue like ICLR.” “The paper does not have sufficient contributions”.
>
> **Authors:**  We would like to humbly disagree with this assessment, at least in part. Our work deals with an important application of machine learning. As such, its objectives are different to that of a methodology focused machine learning work. This is also the reason for our submission to the area “Machine Learning for Sciences (e.g. biology, physics, health sciences, social sciences, climate/sustainability”. Further, to our knowledge machine learning application papers also fall within its scope as indicated in ICLR’s call of papers: “We consider a broad range of subject areas including feature learning, … **as well as applications in** vision, audio, speech , language, music, robotics, games, **healthcare, biology**, sustainability, economics, ethical considerations in ML, and others.”. To the wider ICLR community, it presents an existing challenging and significant problem and proposes using machine learning advancements in semi-supervised learning to approach it.
>
> Our method starts with an assumption regarding the possibility of improvement in performance with the inclusion of this regularization method. We verify this assumption empirically throughout the paper. We think that proposing online generation of mixtures of simulated and real bulk gene expression data, and regularizing the training with a consistency loss is a novel contribution in the task of cell deconvolution. It is an attempt to incorporate information from bulk RNA-seq data in a task dependent manner which has not been explored before. Further, deconvolution is not a task limited to bulk RNA-seq. It arises in several other tasks in cell biology, namely finding origin of cells in DNA methylation, recovering cell type distribution in spatial transcriptomics, signature enrichment for omics e.g. proteomics, microRNA, metabolomics etc. Beyond omics, deconvolution falls in the broader task of semi-blind source separation. Our work demonstrates how utilizing domain knowledge in this context could improve model performances. Depending on the field, the consistency loss may take different forms, or another form of source-mixture simulations would be needed.
>
> -----
>
> **(2) Reviewer:** “Is there sample-to-sample correspondence between $B_i$ and $B_i^\text{sim}$?”
>
> **Authors:** $\mathbf{B}_i$ and $\mathbf{B}_i^{\text{sim}}$ do not have sample correspondence. We used the same index for clarity as indicated in the beginning of Section 2.1.2 “To simplify the notation, we use the same index $i$ for real bulk samples, simulations (sim) and their mixtures (mix, defined further).
> However, following your question, we think that this could be made clearer. For this, we have modified the aforementioned text to now read, “Consider $\mathbf{B}$ represents gene expression matrices of real (\textit{i.e.} test) bulk RNA-seq that we want to deconvolve and and $\mathbf{B}^{\text{sim}}$ represents gene expression matrix of simulated bulk samples. The number of rows, representing samples, in these two matrices may differ. However, to simplify the notation, we use the same index $i$ for real bulk samples, simulations ($\text{sim}$) and their mixtures ($\text{mix}$, defined further).”

---

> > ### Author Response · Authors · 2022-11-18
> > **our response (2 of 4)**
> >
> > **(3) Reviewer**: “There is no justification for the consistency term and it is not clear to me why such a simple linear formulation can improve cell type fraction approximation.”
> > “While bulk measurements and simulated dataset (based on single-cell measurements) consist of different sources of noise and biological variabilities, can you justify why a linear mixture of these two cell measurements can enhance cell deconvolution?”
> >
> > **Authors**: The consistency term is based on an assumption we made in Section 2.1.1, “From Section 2, it is evident that the relationship between B and S is linear. However, S is unobserved and learning is done using simulations. To address the inherent domain shift, we hypothesize that a consistency based regularization penalizing non-linearity of mixtures of real and simulated samples would result in a mapping f that is closer to true f.“ The reasoning of this assumption is based on the linear nature of cell deconvolution. Of course one question that could arise is if the deconvolution task is linear, a linear model will be sufficient. Indeed, if we have access to signatures corresponding to real bulk samples, a linear model may be sufficient. To verify this, we also included linear MLPs in our evaluation. As indicated in Section 4.1, a linear model trained on simulations leads to much higher errors on the real data.
> >
> > The question of “why” is very important but one that is difficult to answer for us with certainty. A straightforward explanation, albeit speculative, would be that cell type-specific genes that characterize a cell type remain invariant across data conditions and modalities (Dominguez Conde et al., 2022), which we note after Eq. 3. Indeed, Scaden, TAPE-O, or linear MLPs all provide non random correlations while they are trained on linearly mixed simulated datasets without utilizing real bulk RNAseq data at all. We reason that consistency regularization improves over existing approaches because the model learns better representations, or in other words, more cell type specific marker genes contribute to the output fractions of that cell type.
> >
> > However, following your very valid question, we looked into possible explanations of our estimates. To this end, we utilized the GradientExplainer method from SHapley Additive exPlanations (SHAP) (Sundararajan et. al., 2017; Lundberg et. al., 2017), which approximates additive feature (in our case, gene) contributions towards the output. On the inferred gene set, we selected the top 50 genes for each cell type and performed cell type enrichment using a large scale single-cell atlas (PanglaoDB included in Enrichr - https://maayanlab.cloud/Enrichr/). We also included Scaden as a comparison. Results are included anonymously here: https://anonymousworkspace.notion.site/Rebuttal-SHAP-on-Scaden-and-DISSECT-59d9cdcd3e2544539086814e7d55c79a. This gene set enrichment computes significance of the gene set selected from SHAP with the gene set that is deemed to be cell type-specific based on the single-cell atlas. Therefore, a high score indicates a better overlap. As clear from the results, both methods identify gene sets that are enriched in the correct cell type. However, for example for CD4 T cells, more CD4 T cells related gene sets are significant and with higher scores for DISSECT. This provides a nice way to approximately verify our assumption that DISSECT focusses more on cell type-specific genes.
> >
> > We would like to point out that SHAP values are not exact. Therefore, we can’t be certain but it provides a nice way to approximately verify.
> >
> > Sundararajan, Mukund, Ankur Taly, and Qiqi Yan. "Axiomatic attribution for deep networks." International conference on machine learning. PMLR, 2017.
> > Lundberg, Scott M., and Su-In Lee. "A unified approach to interpreting model predictions." Advances in neural information processing systems 30 (2017).
> > Domínguez Conde, C., et al. "Cross-tissue immune cell analysis reveals tissue-specific features in humans." Science 376.6594 (2022): eabl5197.
> >
> > -----
> >
> > **(4) Reviewer:** “Section 2.1.2, Eq. 4 should be written as $x_is_i$, not $s_ix_i$.”
> >
> > **Authors**: Thank you for noticing our mistake. We have corrected it in the revised version.
> >
> > -----
> >
> > **(5) Reviewer:** I think reporting MSE (which is reported in the supplement) is more informative rather than reporting correlation between estimated variables and true ones. My suggestion is to report MSE results in the main text.
> >
> > **Authors**: Thank you for this suggestion. We have moved the RMSE table to the main text. In turn we had to move the granular cell type experiment (both correlation and RMSE) to the appendix due to page limitations.

---

> > > ### Author Response · Authors · 2022-11-18
> > > **our response (3 of 4)**
> > >
> > > **(6) Reviewer:** “In Section 2, 4th equation, why does the signature matrix which is a c x n matrix is a function of three indices, including sample id? Does not S represent the expression profiles across cell types?”
> > >
> > > **Authors**: $\mathbf{S}$ represents the expression profiles across cell types per sample. We have indicated this while introducing the notation in Section 2 “In this case, since each simulated sample has a distinct signature (i.e. gene expression profile), $\mathbf{S}$ is a three dimensional matrix with each element $\mathbf{S}_{kji}$ denoting gene expression of gene $j$ in cell type $k$ for sample $i$.“ We have also compared DISSECT with CibersortX and TAPE, both of which measure cell type-specific gene expression per sample.
> > > To make it clearer, we have modified the title of Section 2.2 to now read “Estimation of per sample cell type-specific gene expression profiles”. We have also modified the text in Section 2.2.2 to now read “However, cell type gene expression profiles (at least for genes that are not invariant across tissue states) may differ between samples. Previously, works such as CSx (Newman et al., 2019) and TAPE (Chen et al., 2021) have explored utilizing cell type fractions to estimate gene expression per sample.”
> > >
> > > -----
> > >
> > > **(7) Reviewer:** Is the number of cell types given to the model apriori? How does the deconvolution performance change in the absence of having the true number of cell types in the bulk RNA-seq data?
> > >
> > > **Authors**: The number of cell types given to the model is determined based on the tissue and the cell types of interest. It is possible that the bulk and single cell datasets contain some cell type differences, which we label as unknown in the case of bulk data (we always know all cell types in the single cell data, of course), as outlined in the description of Table 1. This approach works nicely, as already shown in a previous work Scaden compared within our study. To further clarify this point we have modified our submission in Section 4.1:
> > > “Since the number of cell types is unknown a priori in a bulk RNA-seq dataset that we want to deconvolve, we create an ”unknown” cell type label in the reference dataset by merging cells not belonging to any of the cell types present in corresponding tissue Menden et al. (2020); Chen et al. (2021). This unknown cluster allows comparison of fractions measured at relative scale. PBMCs consist of five main cell types namely CD4 T cells, CD8 T cells, NK cells, Monocytes and B cells (Bittersohl et. al., 2016). Thereby, we end up with six cell types (including unknown cell type). Similarly, for the bulk RNA-seq datasets (Table 1), we grouped the ground truth cell type proportions not belonging to these five cell types in a single label “Unknown” following the methodology in Menden et. al. (2020).
> > > To preprocess single-cell datasets, we utilized the procedure described in Appendix C which includes quality control (QC) and simulations. A detailed information on the parameters used for simulations are provided in Appendix C.2.”
> > >
> > > -----
> > >
> > > **(8) Reviewer**: “In Figure 1.a, I think the first “+” operation should be “=”.”
> > >
> > > **Authors**: In Figure 1.A, we illustrated the results of the simulation process for one sample. The simulation process results in a simulated bulk sample with paired cell type fractions and cell type-specific gene expression. We have updated Figure 1 to illustrate the Equation 1, and as such we have replaced first + with = and second + with x. It now reflects B=XS. We would like to thank the reviewer for this great advice to improve the figure.
> > >
> > > -----
> > >
> > > **(9) Reviewer:** “Why do you use a variational approach to learn the signature matrix? Why not just an MLP, similar to X matrix estimation? Is there any plan to use the latent factors?”
> > >
> > > **Authors:** This is a very good comment and we have thought about this quite a bit. While it is possible to use a simple MLP, it would considerably increase the model size and therefore the parameter space to optimize, since we would have to train  c distinct MLP, where c is the number of cell types. The usage of a conditional variational AE allows us to conditionally reconstruct the cell type-specific gene expression in a more effective and elegant manner. We now highlight this  reasoning in the revised Section 2.2.2,
> > > “To jointly train the network on all cell types, we condition the decoder (at its input layer) with cell type labels. This allows for training a single model to estimate gene expression of each cell type for a sample..” Further, the conditional variational framework has already been successfully utilized within genomics (Marouf et. al., 2020) representing a natural choice for us.
> > >
> > > Marouf, Mohamed, et al. "Realistic in silico generation and augmentation of single-cell RNA-seq data using generative adversarial networks." Nature communications 11.1 (2020): 1-12.

---

> > > > ### Author Response · Authors · 2022-11-18
> > > > **our response (4 of 4)**
> > > >
> > > > **(10) Reviewer:** “How many genes are profiled in datasets in Tables 1 & 2? Are you using the same subset of genes in Eq. 3?”
> > > >
> > > > **Authors:** This is a very good question, which obviously hasn't been explained in enough detail.
> > > >
> > > > In the revised manuscript, we have now added the number of genes for each single-cell dataset. We also highlight the number of genes for each bulk dataset in Table 1, now. Since the number of genes differ before and after quality control, we included both pre-QC and post-QC numbers of genes. Further, following the comments from reviewer fXFv, we modified the language of this Section to improve readability.
> > > >
> > > > Geneset between simulated and real datasets may vary between different experiments. To be able to train our model, we used the genes that are common between the reference single-cell dataset (constructed from Table 1) and real bulk RNA-seq (from Table 2). We have added the size of these common genesets for each bulk RNA-seq in the text in Section 4.1. Further, to make it clearer, we have modified to read “Given a true bulk RNA-seq ... ... defined over common gene-set” before Eq. 3.
> > > >
> > > > -----
> > > >
> > > > We hope that these answers satisfied most of your concerns. We welcome and highly value your feedback and comments.

---

> > ### Author Response · Authors · 2022-12-06
> > **Follow-up**
> >
> > We thank the reviewer once again for the feedback. We are looking forward to the assessment of our responses. We would be happy to continue the discussion.

---

### Author Response · Authors · 2022-11-18
**Updated manuscript and responses**

We have now updated our manuscript to incorporate suggestions from the reviewers. We have replied to each reviewer to address their concerns. Below is a summary of what has been changed along with the reasons for those changes.

1. Following Reviewer W86N's concerns and suggestion -

The text while introducing assumption in Section 2 has been expanded. RMSE table from Appendix is moved to the main text and correlation table for estimation of granular level cell type fractions has been moved to Appendix.

Figure 1 is modified to incorporate the reviewer's comment on it.

Number of genes are added in Tables 1 and 2 as well as the common gene set between reference scRNA-seq and bulk RNA-seq in Section 4.

It is made clear that the signature matrix is three dimensional.

2. Following Reviewer w6RR's feedback -

Appendix F is added to evaluate cell type specific gene expression on spatial transcriptomics and referenced in the Section 6.

Comparison with simulation-phase only is added in Appendix G.

3. Following Reviewer qKLA's feedback -

Appendix E has been modified and two additional experiments are added in Appendix E.1.

We hope that our responses have successfully addressed concerns of the reviewers. We look forward to further feedback and discussions.

---

### Decision · Program_Chairs · 2023-01-20

**Decision:**

Reject

**Justification For Why Not Higher Score:**

All reviews recommend rejection of the paper.

**Justification For Why Not Lower Score:**

N/A

**Metareview: Summary, Strengths And Weaknesses:**

The authors propose a new model for cell deconvolution task by introducing consistency regularisation.
The main weaknesses raised by the reviewers is the lack of novelty. There is limited technical novelty. The application, while relevant, is also not a novel ML application, but many related ML methods exist.